# NeuS: Learning Neural Implicit Surfaces by Volume Rendering for Multi-view Reconstruction

**Peng Wang**[†], **Lingjie Liu**[‡*], **Yuan Liu**[†], **Christian Theobalt**[‡], **Taku Komura**[†], **Wenping Wang**[◇*]
[†]The University of Hong Kong  [‡]Max Planck Institute for Informatics
[◇]Texas A&M University
[†]{pwang3,yliu,taku}@cs.hku.hk  [‡]{lliu,theobalt}@mpi-inf.mpg.de
[◇]wenping@tamu.edu

## Abstract

We present a novel neural surface reconstruction method, called *NeuS*, for reconstructing objects and scenes with high fidelity from 2D image inputs. Existing neural surface reconstruction approaches, such as DVR [Niemeyer et al., 2020] and IDR [Yariv et al., 2020], require foreground mask as supervision, easily get trapped in local minima, and therefore struggle with the reconstruction of objects with severe self-occlusion or thin structures. Meanwhile, recent neural methods for novel view synthesis, such as NeRF [Mildenhall et al., 2020] and its variants, use volume rendering to produce a neural scene representation with robustness of optimization, even for highly complex objects. However, extracting high-quality surfaces from this learned implicit representation is difficult because there are not sufficient surface constraints in the representation. In NeuS, we propose to represent a surface as the zero-level set of a *signed distance function* (SDF) and develop a new volume rendering method to train a neural SDF representation. We observe that the conventional volume rendering method causes inherent geometric errors (i.e. bias) for surface reconstruction, and therefore propose a new formulation that is free of bias in the first order of approximation, thus leading to more accurate surface reconstruction even without the mask supervision. Experiments on the DTU dataset and the BlendedMVS dataset show that NeuS outperforms the state-of-the-arts in high-quality surface reconstruction, especially for objects and scenes with complex structures and self-occlusion.

## 1 Introduction

Reconstructing surfaces from multi-view images is a fundamental problem in computer vision and computer graphics. 3D reconstruction with neural implicit representations has recently become a highly promising alternative to classical reconstruction approaches [35, 8, 2] due to its high reconstruction quality and its potential to reconstruct complex objects that are difficult for classical approaches, such as non-Lambertian surfaces and thin structures. Recent works represent surfaces as signed distance functions (SDF) [46, 49, 17, 22] or occupancy [29, 30]. To train their neural models, these methods use a differentiable surface rendering method to render a 3D object into images and compare them against input images for supervision. For example, IDR [46] produces impressive reconstruction results, but it fails to reconstruct objects with complex structures that causes abrupt depth changes. The cause of this limitation is that the surface rendering method used in IDR only considers a single surface intersection point for each ray. Consequently, the gradient only exists at this single point, which is too local for effective back propagation and would get optimization stuck in a poor local minimum when there are abrupt changes of depth on images. Furthermore, object

---

[*]Corresponding authors.

35th Conference on Neural Information Processing Systems (NeurIPS 2021).

masks are needed as supervision for converging to a valid surface. As illustrated in Fig. 1 (a) top, with the radical depth change caused by the hole, the neural network would incorrectly predict the points near the front surface to be blue, failing to find the far-back blue surface. The actual test example in Fig. 1 (b) shows that IDR fails to correctly reconstruct the surfaces near the edges with abrupt depth changes.

Recently, NeRF [28] and its variants have explored to use a volume rendering method to learn a volumetric radiance field for novel view synthesis. This volume rendering approach samples multiple points along each ray and perform $\alpha$-composition of the colors of the sampled points to produce the output pixel colors for training purposes. The advantage of the volume rendering approach is that it can handle abrupt depth changes, because it considers multiple points along the ray and so all the sample points, either near the surface or on the far surface, produce gradient signals for back propagation. For example, referring Fig. 1 (a) bottom, when the near surface (yellow) is found to have inconsistent colors with the input image, the volume rendering approach is capable of training the network to find the far-back surface to produce the correct scene representation. However, since it is intended for novel view synthesis rather than surface reconstruction, NeRF only learns a volume density field, from which it is difficult to extract a high-quality surface. Fig. 1 (b) shows a surface extracted as a level-set surface of the density field learned by NeRF. Although the surface correctly accounts for abrupt depth changes, it contains conspicuous noise in some planar regions.

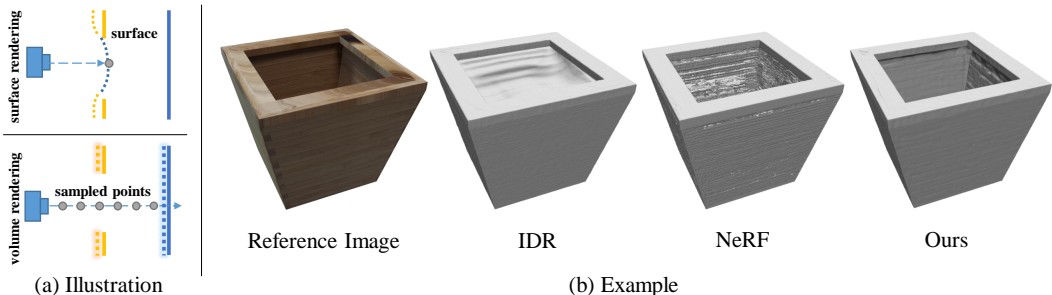

| (a) Illustration | (b) Example |

Figure 1: (a) Illustration of the surface rendering and volume rendering. (b) A toy example of bamboo planter, where there are occlusions on the top of the planter. Compared to the state-of-the-art methods, our approach can handle the occlusions and achieve better reconstruction quality.

In this work, we present a new neural rendering scheme, called *NeuS*, for multi-view surface reconstruction. NeuS uses the *signed distance function* (SDF) for surface representation and uses a novel volume rendering scheme to learn a neural SDF representation. Specifically, by introducing a density distribution induced by SDF, we make it possible to apply the volume rendering approach to learning an implicit SDF representation and thus have the best of both worlds, i.e. an accurate surface representation using a neural SDF model and robust network training in the presence of abrupt depth changes as enabled by volume rendering. Note that simply applying a standard volume rendering method to the density associated with SDF would lead to discernible bias (i.e. inherent geometric errors) in the reconstructed surfaces. This is a new and important observation that we will elaborate later. Therefore we propose a novel volume rendering scheme to ensure unbiased surface reconstruction in the first-order approximation of SDF. Experiments on both DTU dataset and BlendedMVS dataset demonstrated that NeuS is capable of reconstructing complex 3D objects and scenes with severe occlusions and delicate structures, even without foreground masks as supervision. It outperforms the state-of-the-art neural scene representation methods, namely IDR [46] and NeRF [28], in terms of reconstruction quality.

## 2   Related Works

**Classical Multi-view Surface and Volumetric Reconstruction.**   Traditional multi-view 3D reconstruction methods can be roughly classified into two categories: point- and surface-based reconstruction [2, 8, 9, 35] and volumetric reconstruction [6, 3, 36]. Point- and surface-based reconstruction methods estimate the depth map of each pixel by exploiting inter-image photometric consistency [8] and then fuse the depth maps into a global dense point cloud [25, 48]. The surface reconstruction is usually done as a post processing with methods like screened Poisson surface reconstruction [16]. The

reconstruction quality heavily relies on the quality of correspondence matching, and the difficulties in matching correspondence for objects without rich textures often lead to severe artifacts and missing parts in the reconstruction results. Alternatively, volumetric reconstruction methods circumvent the difficulty of explicit correspondence matching by estimating occupancy and color in a voxel grid from multi-view images and evaluating the color consistency of each voxel. Due to limited achievable voxel resolution, these methods cannot achieve high accuracy.

**Neural Implicit Representation.** Some methods enforce 3D understanding in a deep learning framework by introducing inductive biases. These inductive biases can be explicit representations, such as voxel grids [13, 5, 44], point cloud [7, 24, 18], meshes [41, 43, 14], and implicit representations. The implicit representations encoded by a neural network has gained a lot of attention recently, since it is continuous and can achieve high spatial resolution. This representation has been applied successfully to shape representation [26, 27, 31, 4, 1, 10, 47, 32], novel view synthesis [38, 23, 15, 28, 21, 33, 34, 40, 37] and multi-view 3D reconstruction [46, 29, 17, 12, 22].

Our work mainly focuses on learning implicit neural representation encoding both geometry and appearance in 3D space from 2D images via classical rendering techniques. Limited in this scope, the related works can be roughly categorized based on the rendering techniques used, i.e. surface rendering based methods and volume rendering based methods. Surface rendering based methods [29, 17, 46, 22] assume that the color of ray only relies on the color of an intersection of the ray with the scene geometry, which makes the gradient only backpropagated to a local region near the intersection. Therefore, such methods struggle with reconstructing complex objects with severe self-occlusions and sudden depth changes. Furthermore, they usually require object masks as supervision. On the contrary, our method performs well for such challenging cases without the need of masks.

Volume rendering based methods, such as NeRF[28], render an image by $\alpha$-compositing colors of the sampled points along each ray. As explained in the introduction, it can handle sudden depth changes and synthesize high-quality images. However, extracting high-fidelity surface from the learned implicit field is difficult because the density-based scene representation lacks sufficient constraints on its level sets. In contrast, our method combines the advantages of surface rendering based and volume rendering based methods by constraining the scene space as a signed distance function but applying volume rendering to train this representation with robustness. UNISURF [30], a concurrent work, also learns an implicit surface via volume rendering. It improves the reconstruction quality by shrinking the sample region of volume rendering during the optimization. Our method differs from UNISURF in that UNISURF represents the surface by occupancy values, while our method represents the scene by an SDF and thus can naturally extract the surface as the zero-level set of it, yielding better reconstruction accuracy than UNISURF, as will be seen later in the experiment section.

# 3 Method

Given a set of posed images $\{\mathcal{I}_k\}$ of a 3D object, our goal is to reconstruct the surface $\mathcal{S}$ of it. The surface is represented by the zero-level set of a neural implicit SDF. In order to learn the weights of the neural network, we developed a novel volume rendering method to render images from the implicit SDF and minimize the difference between the rendered images and the input images. This volume rendering approach ensures robust optimization in NeuS for reconstructing objects of complex structures.

## 3.1 Rendering Procedure

**Scene representation.** With NeuS, the scene of an object to be reconstructed is represented by two functions: $f : \mathbb{R}^3 \to \mathbb{R}$ that maps a spatial position $\mathbf{x} \in \mathbb{R}^3$ to its signed distance to the object, and $c : \mathbb{R}^3 \times \mathbb{S}^2 \to \mathbb{R}^3$ that encodes the color associated with a point $\mathbf{x} \in \mathbb{R}^3$ and a viewing direction $\mathbf{v} \in \mathbb{S}^2$. Both functions are encoded by Multi-layer Perceptrons (MLP). The surface $\mathcal{S}$ of the object is represented by the zero-level set of its SDF, that is,

$$\mathcal{S} = \left\{ \mathbf{x} \in \mathbb{R}^3 | f(\mathbf{x}) = 0 \right\}. \tag{1}$$

In order to apply a volume rendering method to training the SDF network, we first introduce a probability density function $\phi_s(f(\mathbf{x}))$, called *S-density*, where $f(\mathbf{x})$, $\mathbf{x} \in \mathbb{R}^3$, is the signed distance

function and $\phi_s(x) = se^{-sx}/(1 + e^{-sx})^2$, commonly known as the *logistic density distribution*, is the derivative of the Sigmoid function $\Phi_s(x) = (1 + e^{-sx})^{-1}$, i.e., $\phi_s(x) = \Phi'_s(x)$. In principle $\phi_s(x)$ can be any unimodal (i.e. bell-shaped) density distribution centered at 0; here we choose the logistic density distribution for its computational convenience. Note that the standard deviation of $\phi_s(x)$ is given by $1/s$, which is also a trainable parameter, that is, $1/s$ approaches to zero as the network training converges.

Intuitively, the main idea of NeuS is that, with the aid of the S-density field $\phi_s(f(\mathbf{x}))$, volume rendering is used to train the SDF network with only 2D input images as supervision. Upon successful minimization of a loss function based on this supervision, the zero-level set of the network-encoded SDF is expected to represent an accurately reconstructed surface $\mathcal{S}$, with its induced S-density $\phi_s(f(\mathbf{x}))$ assuming prominently high values near the surface.

**Rendering.** To learn the parameters of the neural SDF and color field, we advise a volume rendering scheme to render images from the proposed SDF representation. Given a pixel, we denote the ray emitted from this pixel as $\{\mathbf{p}(t) = \mathbf{o} + t\mathbf{v} | t \geq 0\}$, where $\mathbf{o}$ is the center of the camera and $\mathbf{v}$ is the unit direction vector of the ray. We accumulate the colors along the ray by

$$C(\mathbf{o}, \mathbf{v}) = \int_0^{+\infty} w(t)c(\mathbf{p}(t), \mathbf{v})\mathrm{d}t, \tag{2}$$

where $C(\mathbf{o}, \mathbf{v})$ is the output color for this pixel, $w(t)$ a weight for the point $\mathbf{p}(t)$, and $c(\mathbf{p}(t), \mathbf{v})$ the color at the point $\mathbf{p}$ along the viewing direction $\mathbf{v}$.

**Requirements on weight function**. The key to learn an accurate SDF representation from 2D images is to build an appropriate connection between output colors and SDF, i.e., to derive an appropriate weight function $w(t)$ on the ray based on the SDF $f$ of the scene. In the following, we list the requirements on the weight function $w(t)$.

1. **Unbiased**. Given a camera ray $\mathbf{p}(t)$, $w(t)$ attains a locally maximal value at a surface intersection point $\mathbf{p}(t^*)$, i.e. with $f(\mathbf{p}(t^*)) = 0$, that is, the point $\mathbf{p}(t^*)$ is on the zero-level set of the SDF $(\mathbf{x})$.

2. **Occlusion-aware**. Given any two depth values $t_0$ and $t_1$ satisfying $f(t_0) = f(t_1)$, $w(t_0) > 0$, $w(t_1) > 0$, and $t_0 < t_1$, there is $w(t_0) > w(t_1)$. That is, when two points have the same SDF value (thus the same SDF-induced S-density value), the point nearer to the view point should have a larger contribution to the final output color than does the other point.

An unbiased weight function $w(t)$ guarantees that the intersection of the camera ray with the zero-level set of SDF contributes most to the pixel color. The occlusion-aware property ensures that when a ray sequentially passes multiple surfaces, the rendering procedure will correctly use the color of the surface nearest to the camera to compute the output color.

Next, we will first introduce a naive way of defining the weight function $w(t)$ that directly using the standard pipeline of volume rendering, and explain why it is not appropriate for reconstruction before introducing our novel construction of $w(t)$.

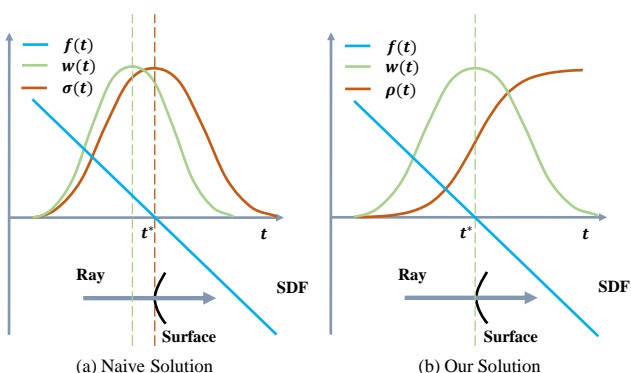

Figure 2: Illustration of (a) weight bias of naive solution, and (b) the weight function defined in our solution, which is unbiased in the first-order approximation of SDF.

**Naive solution**. To make the weight function occlusion-aware, a natural solution is based on the standard volume rendering formulation [28] which defines the weight function by

$$w(t) = T(t)\sigma(t), \tag{3}$$

where $\sigma(t)$ is the so-called *volume density* in classical volume rendering and $T(t) = \exp(-\int_0^t \sigma(u)\mathrm{d}u)$ here denotes the *accumulated transmittance* along the ray. To adopt the stan-

dard volume density formulation [28], here $\sigma(t)$ is set to be equal to the S-density value, i.e. $\sigma(t) = \phi_s(f(\mathbf{p}(t)))$ and the weight function $w(t)$ is computed by Eqn. 3. Although the resulting weight function is occlusion-aware, it is biased as it introduces inherent errors in the reconstructed surfaces. As illustrated in Fig. 2 (a), the weight function $w(t)$ attains a local maximum at a point before the ray reaches the surface point $\mathbf{p}(t^*)$, satisfying $f(\mathbf{p}(t^*)) = 0$. This fact will be proved in the supplementary material.

**Our solution**. To introduce our solution, we first introduce a straightforward way to construct an unbiased weight function, which directly uses the normalized S-density as weights

$$w(t) = \frac{\phi_s(f(\mathbf{p}(t)))}{\int_0^{+\infty} \phi_s(f(\mathbf{p}(u)))\mathrm{d}u}. \tag{4}$$

This construction of weight function is unbiased, but not occlusion-aware. For example, if the ray penetrates two surfaces, the SDF function $f$ will have two zero points on the ray, which leads to two peaks on the weight function $w(t)$ and the resulting weight function will equally blend the colors of two surfaces without considering occlusions.

To this end, now we shall design the weight function $w(t)$ that is both occlusion-aware and unbiased in the first-order approximation of SDF, based on the aforementioned straightforward construction. To ensure an occlusion-aware property of the weight function $w(t)$, we will still follow the basic framework of volume rendering as Eqn. 3. However, different from the conventional treatment as in naive solution above, we define our function $w(t)$ from the S-density in a new manner. We first define an opaque density function $\rho(t)$, which is the counterpart of the volume density $\sigma$ in standard volume rendering. Then we compute the new weight function $w(t)$ by

$$w(t) = T(t)\rho(t), \quad \text{where } T(t) = \exp\left(-\int_0^t \rho(u)\mathrm{d}u\right). \tag{5}$$

**How we derive opaque density $\rho$**. We will first consider a simple case where there is only one surface intersection, and the surface is simply a plane. Since Eqn. 4 indeed satisfies the above requirements under this assumption, we derive the underlying opaque density $\rho$ corresponding to the weight definition of Eqn. 4 using the framework of volume rendering. Then we will generalize this opaque density to the general case of multiple surface intersections.

Specifically, in the simple case of a single plane intersection, it is easy to see that the signed distance function $f(\mathbf{p}(t))$ is $-|\cos(\theta)| \cdot (t - t^*)$, where $f(\mathbf{p}(t^*)) = 0$, and $\theta$ is the angle between the view direction $\mathbf{v}$ and the outward surface normal vector $\mathbf{n}$. Because the surface is assumed locally, $|\cos(\theta)|$ is a constant. It follows from Eqn. 4 that

$$\begin{aligned} w(t) &= \frac{\phi_s(f(\mathbf{p}(t)))}{\int_{-\infty}^{+\infty} \phi_s(f(\mathbf{p}(u)))\mathrm{d}u} \\ &= \frac{\phi_s(f(\mathbf{p}(t)))}{\int_{-\infty}^{+\infty} \phi_s(-|\cos(\theta)| \cdot (u - t^*))\mathrm{d}u} \\ &= \frac{\phi_s(f(\mathbf{p}(t)))}{|\cos(\theta)|^{-1} \cdot \int_{-\infty}^{+\infty} \phi_s(u - t^*)\mathrm{d}u} \\ &= |\cos(\theta)|\phi_s(f(\mathbf{p}(t))). \end{aligned} \tag{6}$$

Recall that the weight function within the framework of volume rendering is given by $w(t) = T(t)\rho(t)$, where $T(t) = \exp(-\int_0^t \rho(u)\mathrm{d}u)$ denotes the *accumulated transmittance*. Therefore, to derive $\rho(t)$, we have

$$T(t)\rho(t) = |\cos(\theta)|\phi_s(f(\mathbf{p}(t))). \tag{7}$$

Since $T(t) = \exp(-\int_0^t \rho(u)\mathrm{d}u)$, it is easy to verify that $T(t)\rho(t) = -\frac{\mathrm{d}T}{\mathrm{d}t}(t)$. Further, note that $|\cos(\theta)|\phi_s(f(\mathbf{p}(t))) = -\frac{\mathrm{d}\Phi_s}{\mathrm{d}t}(f(\mathbf{p}(t)))$. It follows that $\frac{\mathrm{d}T}{\mathrm{d}t}(t) = \frac{\mathrm{d}\Phi_s}{\mathrm{d}t}(f(\mathbf{p}(t)))$. Integrating both sides of this equation yields

$$T(t) = \Phi_s(f(\mathbf{p}(t))). \tag{8}$$

Taking the logarithm and then differentiating both sides, we have

$$\int_{-\infty}^{t} \rho(u)\mathrm{d}u = -\ln(\Phi_s(f(\mathbf{p}(t))))$$

$$\Rightarrow \rho(t) = \frac{-\frac{\mathrm{d}\Phi_s}{\mathrm{d}t}(f(\mathbf{p}(t)))}{\Phi_s(f(\mathbf{p}(t)))}. \tag{9}$$

This is the formula of the opaque density $\rho(t)$ in case of single plane intersection. The weight function $w(t)$ induced by $\rho(t)$ is shown in Figure 2(b). Now we generalize the opaque density to the general case where there are multiple surface intersections along the ray $\mathbf{p}(t)$. In this case, $-\frac{\mathrm{d}\Phi_s}{\mathrm{d}t}(f(\mathbf{p}(t)))$ becomes negative on the segment of the ray with increasing SDF values. Thus we clip it against zero to ensure that the value of $\rho$ is always non-negative. This gives the following opaque density function $\rho(t)$ in general cases.

$$\rho(t) = \max\left(\frac{-\frac{\mathrm{d}\Phi_s}{\mathrm{d}t}(f(\mathbf{p}(t)))}{\Phi_s(f(\mathbf{p}(t)))}, 0\right). \tag{10}$$

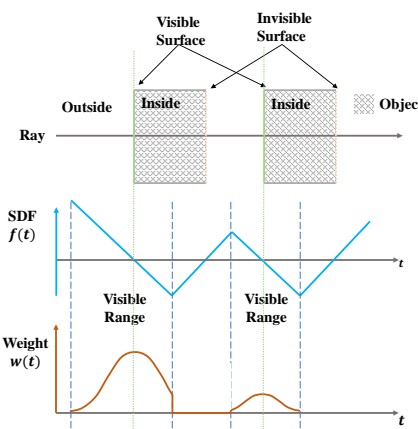

Figure 3: Illustration of weight distribution in case of multiple surface intersection.

Based on this equation, the weight function $w(t)$ can be computed with standard volume rendering as in Eqn. 5. The illustration in the case of multiple surface intersection is shown in Figure 3.

The following theorem states that in general cases (i.e., including both single surface intersection and multiple surface intersections) the weight function defined by Eqn. 10 and Eqn. 5 is unbiased in the first-order approximation of SDF. The proof is given in the supplementary material.

**Theorem 1** *Suppose that a smooth surface $\mathbb{S}$ is defined by the zero-level set of the signed distance function $f(\mathbf{x}) = 0$, and a ray $\mathbf{p}(t) = \mathbf{o} + t\mathbf{v}$ enters the surface $\mathbb{S}$ from outside to inside, with the intersection point at $\mathbf{p}(t^*)$, that is, $f(\mathbf{p}(t^*)) = 0$ and there exists an interval $[t_l, t_r]$ such that $t^* \in [t_l, t_r]$ and $f(\mathbf{p}(t))$ is monotonically decreasing in $[t_l, t_r]$. Suppose that in this local interval $[t_l, t_r]$, the surface can be tangentially approximated by a sufficiently small planar patch, i.e., $\nabla\mathbf{f}$ is regarded as fixed. Then, the weight function $w(t)$ computed by Eqn. 10 and Eqn. 5 in $[t_l, t_r]$ attains its maximum at $t^*$.*

**Discretization**. To obtain discrete counterparts of the opacity and weight function, we adopt the same approximation scheme as used in NeRF [28], This scheme samples $n$ points $\{\mathbf{p}_i = \mathbf{o} + t_i\mathbf{v} | i = 1, ..., n, t_i < t_{i+1}\}$ along the ray to compute the approximate pixel color of the ray as

$$\hat{C} = \sum_{i=1}^{n} T_i\alpha_i c_i, \tag{11}$$

where $T_i$ is the discrete *accumulated transmittance* defined by $T_i = \prod_{j=1}^{i-1}(1 - \alpha_j)$, and $\alpha_i$ is discrete opacity values defined by

$$\alpha_i = 1 - \exp\left(-\int_{t_i}^{t_{i+1}} \rho(t)\mathrm{d}t\right), \tag{12}$$

which can further be shown to be

$$\alpha_i = \max\left(\frac{\Phi_s(f(\mathbf{p}(t_i))) - \Phi_s(f(\mathbf{p}(t_{i+1})))}{\Phi_s(f(\mathbf{p}(t_i)))}, 0\right). \tag{13}$$

The detailed derivation of this formula for $\alpha_i$ is given in the supplementary material.

## 3.2 Training

To train NeuS, we minimize the difference between the rendered colors and the ground truth colors, without any 3D supervision. Besides colors, we can also utilize the masks for supervision if provided.

Specifically, we optimize our neural networks and inverse standard deviation $s$ by randomly sampling a batch of pixels and their corresponding rays in world space $P = \{C_k, M_k, \mathbf{o}_k, \mathbf{v}_k\}$, where $C_k$ is its pixel color and $M_k \in \{0, 1\}$ is its optional mask value, from an image in every iteration. We assume the point sampling size is $n$ and the batch size is $m$. The loss function is defined as

$$\mathcal{L} = \mathcal{L}_{color} + \lambda \mathcal{L}_{reg} + \beta \mathcal{L}_{mask}. \tag{14}$$

The color loss $\mathcal{L}_{color}$ is defined as

$$\mathcal{L}_{color} = \frac{1}{m} \sum_k \mathcal{R}(\hat{C}_k, C_k). \tag{15}$$

Same as IDR[46], we empirically choose $\mathcal{R}$ as L1 loss, which in our observation is robust to outliers and stable in training.

We add an Eikonal term [10] on the sampled points to regularize the SDF of $f_\theta$ by

$$\mathcal{L}_{reg} = \frac{1}{nm} \sum_{k,i} (\|\nabla f(\hat{\mathbf{p}}_{k,i})\|_2 - 1)^2. \tag{16}$$

The optional mask loss $\mathcal{L}_{mask}$ is defined as

$$\mathcal{L}_{mask} = \text{BCE}(M_k, \hat{O}_k), \tag{17}$$

where $\hat{O}_k = \sum_{i=1}^{n} T_{k,i} \alpha_{k,i}$ is the sum of weights along the camera ray, and BCE is the binary cross entropy loss.

**Hierarchical sampling**. In this work, we follow a similar hierarchical sampling strategy as in NeRF [28]. We first uniformly sample the points on the ray and then iteratively conduct importance sampling on top of the coarse probability estimation. The difference is that, unlike NeRF which simultaneously optimizes a coarse network and a fine network, we only maintain one network, where the probability in coarse sampling is computed based on the S-density $\phi_s(f(\mathbf{x}))$ with fixed standard deviations while the probability of fine sampling is computed based on $\phi_s(f(\mathbf{x}))$ with the learned $s$. Details of hierarchical sampling strategy are provided in supplementary materials.

## 4 Experiments

### 4.1 Experimental settings

**Datasets.** To evaluate our approach and baseline methods, we use 15 scenes from the DTU dataset [11], same as those used in IDR [46], with a wide variety of materials, appearance and geometry, including challenging cases for reconstruction algorithms, such as non-Lambertian surfaces and thin structures. Each scene contains 49 or 64 images with the image resolution of $1600 \times 1200$. Each scene was tested with and without foreground masks provided by IDR [46]. We further tested on 7 challenging scenes from the low-res set of the BlendedMVS dataset [45](CC-4 License). Each scene has $31 - 143$ images at $768 \times 576$ pixels and masks are provided by the BlendedMVS dataset. We further captured two thin objects with 32 input images to test our approach on thin structure reconstruction.

**Baselines.** (1) The state-of-the-art surface rendering approach – IDR [46]: IDR can reconstruct surface with high quality but requires foreground masks as supervision; Since IDR has demonstrated superior quality compared to another surface rendering based method – DVR [29], we did not conduct a comparison with DVR. (2) The state-of-the-art volume rendering approach – NeRF [28]: We use a threshold of 25 to extract mesh from the learned density field. We validate this choice in the supplementary material. (3) A widely-used classical MVS method – COLMAP [35]: We reconstruct a mesh from the output point cloud of COLMAP with Screened Poisson Surface Reconstruction [16]. (4) The concurrent work which unifies surface rendering and volume rendering with an occupancy field as scene representation – UNISURF [30]. More details of the baseline methods are included in the supplementary material.

**Implementation details.** We assume the region of interest is inside a unit sphere. We sample 512 rays per batch and train our model for 300k iterations for 14 hours (for the 'w/ mask' setting) and 16 hours (for the 'w/o mask' setting) on a single NVIDIA RTX2080Ti GPU. For the 'w/o mask' setting, we model the background by NeRF++ [50]. Our network architecture and initialization scheme are similar to those of IDR [46]. More details of the network architecture and training parameters can be found in the supplementary material.

| ScanID | w/ mask | | | w/o mask | | | |
|---|---|---|---|---|---|---|---|
| | IDR | NeRF | Ours | COLMAP | NeRF | UNISURF | Ours |
| scan24 | 1.63 | 1.83 | **0.83** | **0.81** | 1.90 | 1.32 | 1.00 |
| scan37 | 1.87 | 2.39 | **0.98** | 2.05 | 1.60 | **1.36** | 1.37 |
| scan40 | 0.63 | 1.79 | **0.56** | **0.73** | 1.85 | 1.72 | 0.93 |
| scan55 | 0.48 | 0.66 | **0.37** | 1.22 | 0.58 | 0.44 | **0.43** |
| scan63 | **1.04** | 1.79 | 1.13 | 1.79 | 2.28 | 1.35 | **1.10** |
| scan65 | 0.79 | 1.44 | **0.59** | 1.58 | 1.27 | 0.79 | **0.65** |
| scan69 | 0.77 | 1.50 | **0.60** | 1.02 | 1.47 | 0.80 | **0.57** |
| scan83 | 1.33 | **1.20** | 1.45 | 3.05 | 1.67 | 1.49 | **1.48** |
| scan97 | 1.16 | 1.96 | **0.95** | 1.40 | 2.05 | 1.37 | **1.09** |
| scan105 | **0.76** | 1.27 | 0.78 | 2.05 | 1.07 | 0.89 | **0.83** |
| scan106 | 0.67 | 1.44 | **0.52** | 1.00 | 0.88 | 0.59 | **0.52** |
| scan110 | **0.90** | 2.61 | 1.43 | 1.32 | 2.53 | 1.47 | **1.20** |
| scan114 | 0.42 | 1.04 | **0.36** | 0.49 | 1.06 | 0.46 | **0.35** |
| scan118 | 0.51 | 1.13 | **0.45** | 0.78 | 1.15 | 0.59 | **0.49** |
| scan122 | 0.53 | 0.99 | **0.45** | 1.17 | 0.96 | 0.62 | **0.54** |
| mean | 0.90 | 1.54 | **0.77** | 1.36 | 1.49 | 1.02 | **0.84** |

Table 1: Quantitative evaluation on DTU dataset. COLMAP results are achieved by trim=0.

## 4.2 Comparisons

We conducted the comparisons in two settings, with mask supervision (w/ mask) and without mask supervision (w/o mask). We measure the reconstruction quality with the Chamfer distances in the same way as UNISURF [30] and IDR [46] and report the scores in Table 1. The results show that our approach outperforms the baseline methods on the DTU dataset in both settings – w/ and w/o mask in terms of the Chamfer distance. Note that the reported scores of IDR in the setting of w/ mask and NeRF and UNISURF in the w/o mask setting are from IDR [46] and UNISURF [30].

We conduct the qualitative comparisons on the DTU dataset and the BlendedMVS dataset in both settings, w/ mask and w/o mask, in Figure 4 and Figure 5, respectively. As shown in Figure 4 for the setting of w/ mask, IDR shows limited performance for reconstructing thin metals parts in Scan 37 (DTU), and fails to handle sudden depth changes in Stone (BlendedMVS) due to the local optimization process in surface rendering. The extracted meshes of NeRF are noisy since the volume density field has not sufficient constraint on its 3D geometry. Regarding the w/o mask setting, we visually compare our method with NeRF and COLMAP in the setting of w/o mask in Figure 5, which shows our reconstructed surfaces are with more fidelity than baselines. We further show a comparison

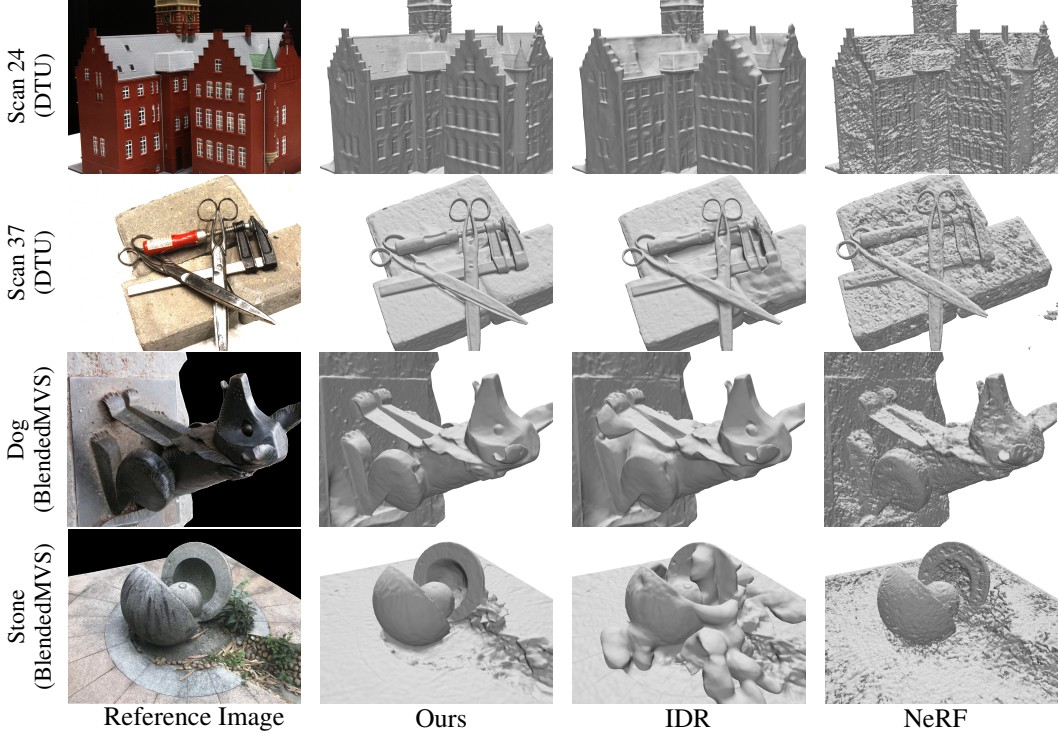

Figure 4: Comparions on surface reconstruction with mask supervision.

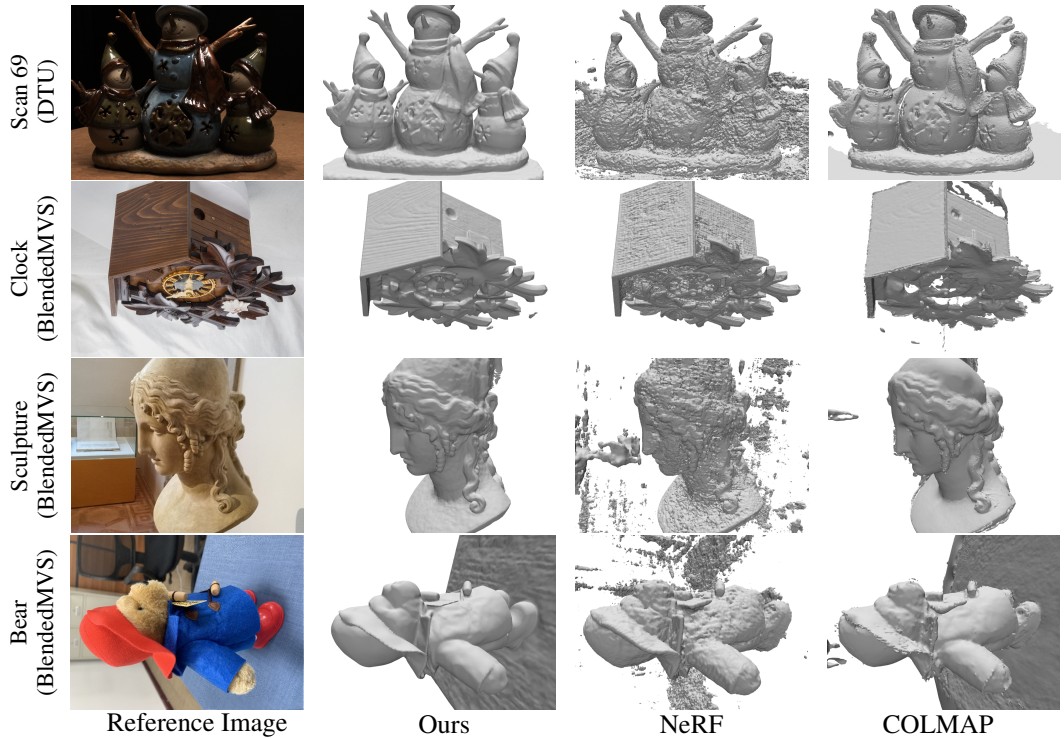

Figure 5: Comparions on surface reconstruction without mask supervision.

with UNISURF [30] on two examples in the w/o mask setting. Note that we use the qualitative results of UNISURF reported their paper for comparison. Our method works better for the objects with abrupt depth changes. More qualitative images are included in the supplementary material.

## 4.3 Analysis

### Ablation study.

To evaluate the effect of the weight calculation, we test three different kinds of weight constructions described in Sec. 3.1: (a) Naive Solution. (b) Straightforward Construction as shown in Eqn. 4. (e) Full Model. As shown in Figure 6, the quantitative result of naive solution is worse than our weight choice (e) in terms of the Chamfer distance. This is because it introduces a bias to the surface reconstruction. If direct construction is used, there are severe artifacts.

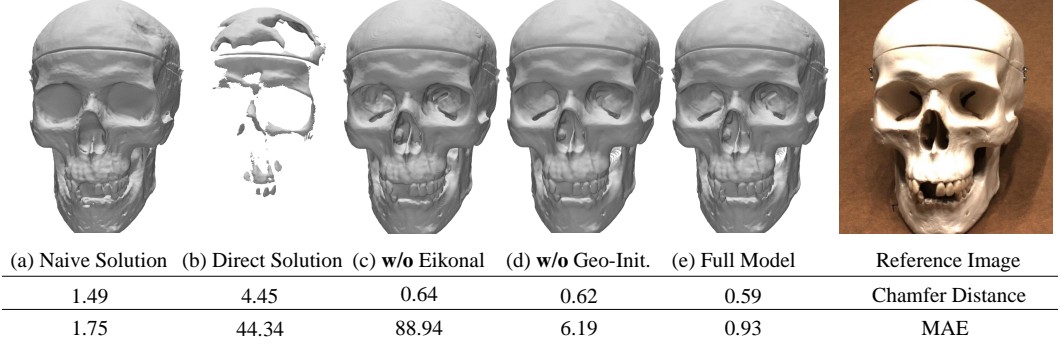

| (a) Naive Solution | (b) Direct Solution | (c) **w/o** Eikonal | (d) **w/o** Geo-Init. | (e) Full Model | Reference Image |
|---|---|---|---|---|---|
| 1.49 | 4.45 | 0.64 | 0.62 | 0.59 | Chamfer Distance |
| 1.75 | 44.34 | 88.94 | 6.19 | 0.93 | MAE |

Figure 6: Ablation studies. We show the qualitative results and report the quantitative metrics in Chamfer distance and MAE (mean absolute error) between the ground-truth and predicted SDF values.

We also studied the effect of Eikonal regularization [10] and geometric initialization [1]. Without Eikonal regularization or geometric initialization, the result on Chamfer distance is on par with that of the full model. However, neither of them can correctly output a signed distance function. This is indicated by the MAE(mean absolute error) between the SDF predictions and corresponding ground-truth SDF, as shown in the bottom line of Figure 6. The MAE is computed on uniformly-sampled points in the object's bounding sphere. Qualitative results of SDF predictions are provided in the supplementary material.

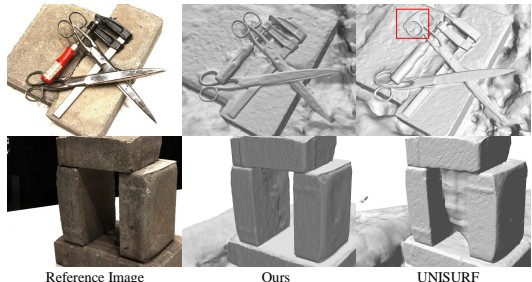

Reference Image     Ours     UNISURF

Figure 7: Visual comparisons with UNISURF.

**Thin structures.** We additionally show results on two challenging thin objects with 32 input images. The plane with rich texture under the object is used for camera calibration. As shown in Fig. 8, our method is able to accurately reconstruct these thin structures, especially on the edges with abrupt depth changes. Furthermore, different from the methods [39, 19, 42, 20] which only target at high-quality thin structure reconstruction, our method can handle the scenes which have a mixture of thin structures and general objects.

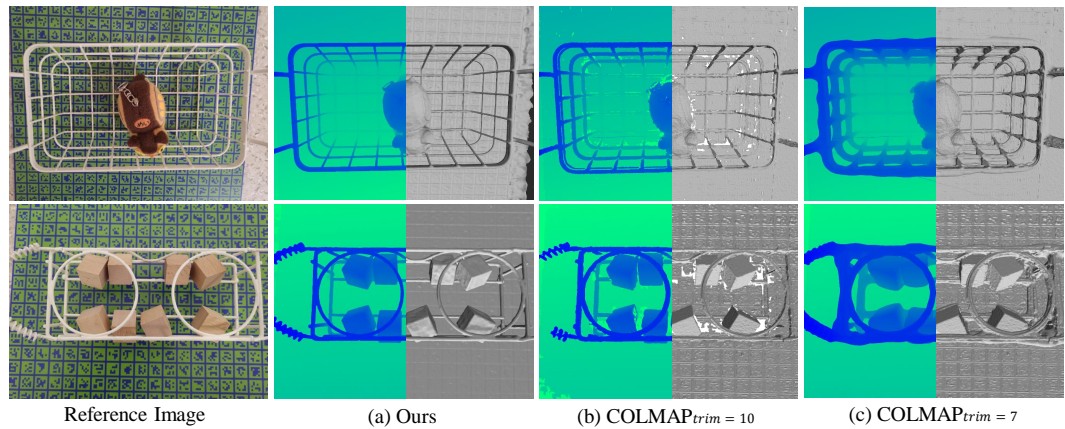

Reference Image     (a) Ours     (b) COLMAP$_{trim = 10}$     (c) COLMAP$_{trim = 7}$

Figure 8: Comparison on scenes with thin structure objects. Left half is the depth map while right half is the reconstructed surface.

# 5 Conclusion

We have proposed *NeuS*, a new approach to multiview surface reconstruction that represents 3D surfaces as neural SDF and developed a new volume rendering method for training the implicit SDF representation. NeuS produces high-quality reconstruction and successfully reconstructs objects with severe occlusions and complex structures. It outperforms the state-of-the-arts both qualitatively and quantitatively. One limitation of our method is that although our method does not heavily rely on correspondence matching of texture features, the performance would still degrade for textureless objects (we show the failure cases in the supplementary material). Moreover, NeuS has only a single scale parameter $s$ that is used to model the standard deviation of the probability distribution for all the spatial location. Hence, an interesting future research topic is to model the probability with different variances for different spatial locations together with the optimization of scene representation, depending on different local geometric characteristics. Negative societal impact: like many other learning-based works, our method requires a large amount of computational resources for network training, which can be a concern for global climate change.

## Acknowlegements

We thank Michael Oechsle for providing the results of UNISURF. Christian Theobalt was supported by ERC Consolidator Grant 770784. Lingjie Liu was supported by Lise Meitner Postdoctoral Fellowship. Computational resources are mainly provided by HKU GPU Farm.

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
