# NeuS: Learning Neural Implicit Surfaces by Volume Rendering for Multi-view Reconstruction - Supplementary Material -

## A    Derivation for Computing Opacity $\alpha_i$

In this section we will derive the formula in Eqn. 13 of the paper for computing the discrete opacity $\alpha_i$. Recall that the opaque density function $\rho(t)$ is defined as

$$
\begin{aligned}
\rho(t) &= \max\left(\frac{-\frac{\mathrm{d}\Phi_s}{\mathrm{d}t}(f(\mathbf{p}(t)))}{\Phi_s(f(\mathbf{p}(t)))}, 0\right) \\
&= \max\left(\frac{-(\nabla f(\mathbf{p}(t)) \cdot \mathbf{v})\phi_s(f(\mathbf{p}(t)))}{\Phi_s(f(\mathbf{p}(t)))}, 0\right),
\end{aligned}
\tag{1}
$$

where $\phi_s(x)$ and $\Phi_s(x)$ are the probability density function (PDF) and cumulative distribution function (CDF) of logistic distribution, respectively. First consider the case where the sample point interval $[t_i, t_{i+1}]$ lies in a range $[t_\ell, t_r]$ over which the camera ray is entering the surface from outside to inside, i.e. the signed distance function is decreasing on the camera ray $\mathbf{p}(t)$ over $[t_\ell, t_r]$. Then it is easy to see that $-(\nabla f(\mathbf{p}(t)) \cdot \mathbf{v}) > 0$ in $[t_i, t_{i+1}]$. It follows from Eqn. 12 of the paper that,

$$
\begin{aligned}
\alpha_i &= 1 - \exp\left(-\int_{t_i}^{t_{i+1}} \rho(t)\mathrm{d}t\right) \\
&= 1 - \exp\left(-\int_{t_i}^{t_{i+1}} \frac{-(\nabla f(\mathbf{p}(t)) \cdot \mathbf{v})\phi_s(f(\mathbf{p}(t)))}{\Phi_s(f(\mathbf{p}(t)))}\mathrm{d}t\right).
\end{aligned}
\tag{2}
$$

Note that the integral term is computed by

$$
\int \frac{-(\nabla f(\mathbf{p}(t)) \cdot \mathbf{v})\phi_s(f(\mathbf{p}(t)))}{\Phi_s(f(\mathbf{p}(t)))}\mathrm{d}t = -\ln(\Phi_s(f(\mathbf{p}(t)))) + C,
\tag{3}
$$

where $C$ is a constant. Thus the discrete opacity can be computed by

$$
\begin{aligned}
\alpha_i &= 1 - \exp\left[-\left(-\ln(\Phi_s(f(\mathbf{p}(t_{i+1})))) + \ln(\Phi_s(f(\mathbf{p}(t_i))))\right)\right] \\
&= 1 - \frac{\Phi_s(f(\mathbf{p}(t_{i+1})))}{\Phi_s(f(\mathbf{p}(t_i)))} \\
&= \frac{\Phi_s(f(\mathbf{p}(t_i))) - \Phi_s(f(\mathbf{p}(t_{i+1})))}{\Phi_s(f(\mathbf{p}(t_i)))}.
\end{aligned}
\tag{4}
$$

Next consider the case where $[t_i, t_{i+1}]$ lies in a range $[t_\ell, t_r]$ over which the camera ray is exiting the surface, i.e. the signed distance function is increasing on $\mathbf{p}(t)$ over $[t_\ell, t_r]$. Then we have $-(\nabla f(\mathbf{p}(t)) \cdot \mathbf{v}) < 0$ in $[t_i, t_{i+1}]$. Then, according to Eqn. 1, we have $\rho(t) = 0$. Therefore, by Eqn. 12 of the paper, we have

$$
\alpha_i = 1 - \exp\left(-\int_{t_i}^{t_{i+1}} \rho(t)\mathrm{d}t\right) = 1 - \exp\left(-\int_{t_i}^{t_{i+1}} 0\mathrm{d}t\right) = 0.
$$

35th Conference on Neural Information Processing Systems (NeurIPS 2021), Sydney, Australia.

Hence, the alpha value $\alpha_i$ in this case is given by

$$\alpha_i = \max\left(\frac{\Phi_s(f(\mathbf{p}(t_i))) - \Phi_s(f(\mathbf{p}(t_{i+1})))}{\Phi_s(f(\mathbf{p}(t_i)))}, 0\right). \tag{5}$$

This completes the derivation of Eqn. 13 of the paper.

## B First-order Bias Analysis

### B.1 Proof of Unbiased Property of Our Solution

PROOF OF THEOREM 1: Suppose that the ray is going from outside to inside of the surface. Hence, we have $-(\nabla f(\mathbf{p}(t)) \cdot \mathbf{v}) > 0$, because by convention the signed distance function $f(\mathbf{x})$ is positive outside and negative inside of the surface.

Recall that our S-density field $\phi_s(f(\mathbf{x}))$ is defined using the logistic density function $\phi_s(x) = se^{-sx}/(1 + e^{-sx})^2$, which is the derivative of the Sigmoid function $\Phi_s(x) = (1 + e^{-sx})^{-1}$, i.e. $\phi_s(x) = \Phi'_s(x)$.

According to Eqn. 5 of the paper, the weight function $w(t)$ is given by

$$w(t) = T(t)\rho(t),$$

where

$$\rho(t) = \max\left(\frac{-(\nabla f(\mathbf{p}(t)) \cdot \mathbf{v})\phi_s(f(\mathbf{p}(t)))}{\Phi_s(f(\mathbf{p}(t)))}, 0\right).$$

By assumption, $-(\nabla f(\mathbf{p}(t)) \cdot \mathbf{v}) > 0$ for $t \in [t_l, t_r]$. Since $\phi_s$ is a probability density function, we have $\phi_s(f(\mathbf{p}(t))) > 0$. Clearly, $\Phi_s(f(\mathbf{p}(t))) > 0$. It follows that

$$\rho(t) = \frac{-(\nabla f(\mathbf{p}(t)) \cdot \mathbf{v})\phi_s(f(\mathbf{p}(t)))}{\Phi_s(f(\mathbf{p}(t)))},$$

which is positive. Hence,

$$
\begin{aligned}
w(t) =& T(t)\rho(t) \\
=& \exp\left(-\int_0^t \rho(t')\mathrm{d}t'\right)\rho(t) \\
=& \exp\left(-\int_0^{t_l} \rho(t')\mathrm{d}t'\right)\exp\left(-\int_{t_l}^t \rho(t')\mathrm{d}t'\right)\rho(t) \\
=& T(t_l)\exp\left(-\int_{t_l}^t \rho(t')\mathrm{d}t'\right)\rho(t) \\
=& T(t_l)\exp\left[-(-\ln(\Phi_s(f(\mathbf{p}(t)))) + \ln(\Phi_s(f(\mathbf{p}(t_l)))))\right]\rho(t) \\
=& T(t_l)\frac{\Phi_s(f(\mathbf{p}(t)))}{\Phi_s(f(\mathbf{p}(t_l)))}\frac{-(\nabla f(\mathbf{p}(t)) \cdot \mathbf{v})\phi_s(f(\mathbf{p}(t)))}{\Phi_s(f(\mathbf{p}(t)))} \\
=& \frac{-(\nabla f(\mathbf{p}(t)) \cdot \mathbf{v})T(t_l)}{\Phi_s(f(\mathbf{p}(t_l)))}\phi_s(f(\mathbf{p}(t))).
\end{aligned} \tag{6}
$$

As a first-order approximation of signed distance function $f$, suppose that locally the surface is tangentially approximated by a sufficiently small planar patch with its outward unit normal vector denoted as $\mathbf{n}$. Because $f(\mathbf{x})$ is a signed distance function, locally it has a unit gradient vector $\nabla f = \mathbf{n}$. Then we have

$$
\begin{aligned}
w(t) =& \frac{-(\nabla f(\mathbf{p}(t)) \cdot \mathbf{v})T(t_l)}{\Phi_s(f(\mathbf{p}(t_l)))}\phi_s(f(\mathbf{p}(t))) \\
=& \frac{|\cos(\theta)|T(t_l)}{\Phi_s(f(\mathbf{p}(t_l)))}\phi_s(f(\mathbf{p}(t))),
\end{aligned} \tag{7}
$$

where $\theta$ is the angle between the view direction $\mathbf{v}$ and the unit normal vector $\mathbf{n}$, that is, $\cos(\theta) = \mathbf{v} \cdot \mathbf{n}$. Here $|\cos(\theta)|T(t_l) \cdot \Phi_s(f(\mathbf{p}(t_l)))^{-1}$ can be regarded as a constant. Hence, $w(t)$ attains a local

maximum when $f(\mathbf{p}(t)) = 0$ because $\phi_s(x)$ is a unimodal density function attaining the maximal value at $x = 0$.

We remark that in this proof we do not make any assumption on the existence of surfaces between the camera and the sample point $\mathbf{p}(t_l)$. Therefore the conclusion holds true for the case of multiple surface intersections on the camera ray. This completes the proof. $\square$

### B.2 Bias in Naive Solution

In this section we show that the weight function derived in naive solution is biased. According to Eqn. 3 of the paper, $w(t) = T(t)\sigma(t)$, with the opacity $\sigma(t) = \phi_s(f(\mathbf{p}(t)))$. Then we have

$$
\begin{aligned}
\frac{\mathrm{d}w}{\mathrm{d}t} &= \frac{\mathrm{d}(T(t)\sigma(t))}{\mathrm{d}t} \\
&= \frac{\mathrm{d}T(t)}{\mathrm{d}t}\sigma(t) + T(t)\frac{\mathrm{d}\sigma(t)}{\mathrm{d}t} \\
&= \left[\exp\left(-\int_0^t \sigma(t)\mathrm{d}t\right)(-\sigma(t))\right]\sigma(t) + T(t)\frac{\mathrm{d}\sigma(t)}{\mathrm{d}t} \\
&= T(t)(-\sigma(t))\sigma(t) + T(t)\frac{\mathrm{d}\sigma(t)}{\mathrm{d}t} \\
&= T(t)\left(\frac{\mathrm{d}\sigma(t)}{\mathrm{d}t} - \sigma(t)^2\right).
\end{aligned}
\tag{8}
$$

Now we perform the same first-order approximation of signed distance function $f$ near the surface intersection as in Section B.1. In this condition, the above equation can be rewritten as

$$
\begin{aligned}
\frac{\mathrm{d}w}{\mathrm{d}t} &= T(t)\left((\nabla f(\mathbf{p}(t)) \cdot \mathbf{v})\phi_s'(f(\mathbf{p}(t))) - \phi_s(f(\mathbf{p}(t)))^2\right) \\
&= T(t)\left(\cos(\theta)\phi_s'(f(\mathbf{p}(t))) - \phi_s(f(\mathbf{p}(t)))^2\right).
\end{aligned}
\tag{9}
$$

Here $\cos(\theta)$ can be regarded as a constant. Now suppose $\mathbf{p}(t^*)$ is a point on the surface $\mathbb{S}$, that is, $f(\mathbf{p}(t^*)) = 0$. Next we will examine the value of $\frac{\mathrm{d}w}{\mathrm{d}t}(t)$ at $t = t^*$. First, clearly, $T(t^*) > 0$ and $\phi_s(f(\mathbf{p}(t^*)))^2 > 0$. Then, since $\phi_s'(0) = 0$, we have

$$
\frac{\mathrm{d}w}{\mathrm{d}t}(t^*) = T(t^*)(\cos(\theta)\phi_s'(0) - \sigma(t^*)^2) = -T(t^*)\phi_s(0)^2 < 0.
$$

Hence $w(t)$ in naive solution does not attain a local maximum at $t = t^*$, which corresponds to a point on the surface $\mathbb{S}$. This completes the proof. $\square$

## C  Second-order Bias Analysis

In this section we briefly introduce our local analysis in the interval $[t_l, t_r]$ near the surface intersection, in second-order approximation. In this condition, we follow the similar assumption as Section B that the signed distance function $f(\mathbf{p}(t))$ monotonically decreases along the ray in the interval $[t_l, t_r]$.

According to Eqn. 8, the derivative of $w(t)$ is given by:

$$
\frac{\mathrm{d}w}{\mathrm{d}t} = T(t)\left(\frac{\mathrm{d}\sigma(t)}{\mathrm{d}t} - \sigma(t)^2\right).
$$

Clearly, we have $T(t) > 0$. Hence, when $w(t)$ attains local maximum at $\bar{t}$, there is $\left(\frac{\mathrm{d}\sigma(\bar{t})}{\mathrm{d}t} - \sigma(\bar{t})^2\right) = 0$.

**The case of our solution**. In our solution, the volume density is given by $\sigma(t) = \rho(t)$ following Eqn. 1. After organizing, we have

$$
\frac{\mathrm{d}^2 f}{\mathrm{d}t^2}(\mathbf{p}(\bar{t})) \cdot \phi_s(f(\mathbf{p}(\bar{t}))) + \left(\frac{\mathrm{d}f}{\mathrm{d}t}(\mathbf{p}(\bar{t}))\right)^2 \phi_s'(f(\mathbf{p}(\bar{t}))) = 0.
$$

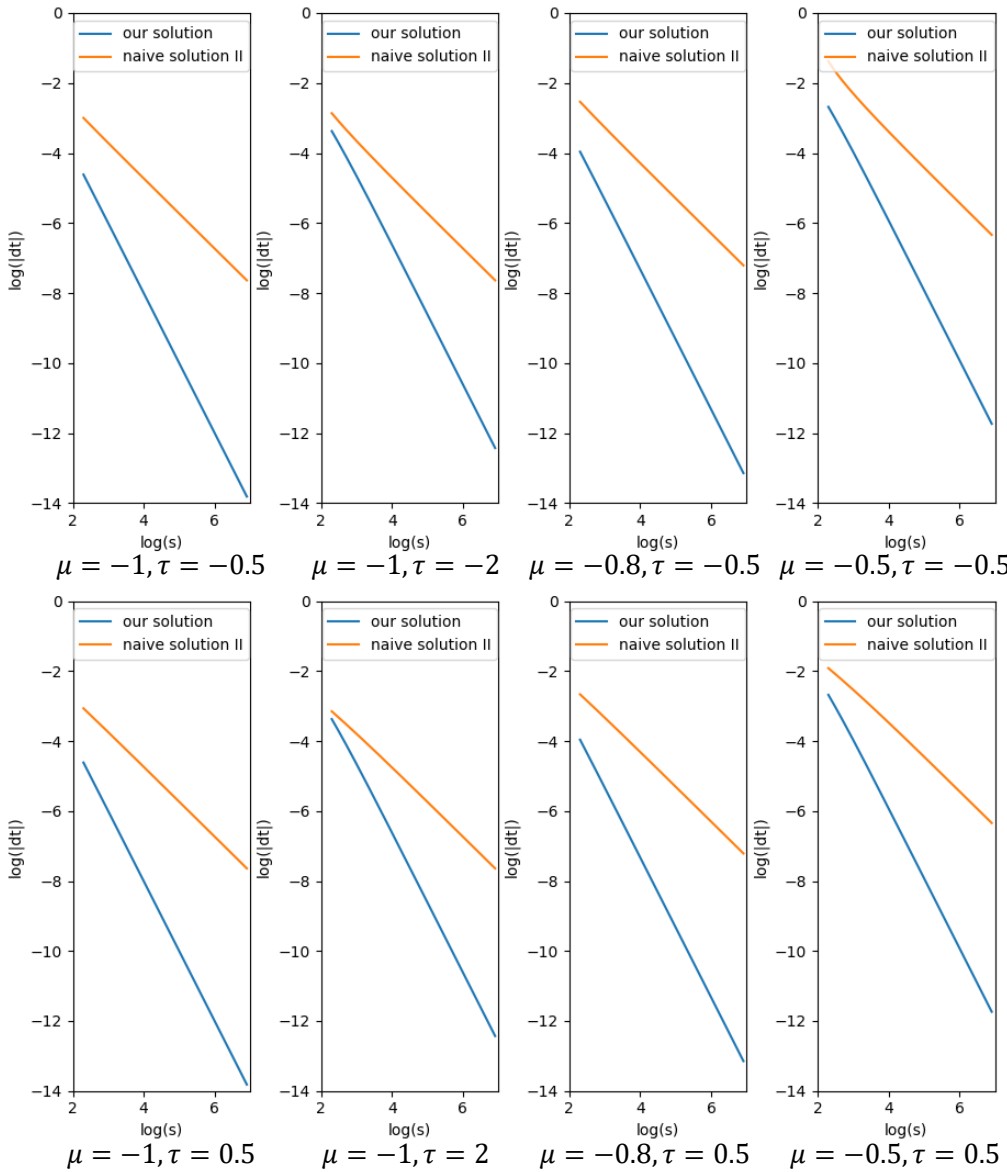

Figure 1: The curve of $\Delta_t$ versus $s$, given fixed $\mu, \tau$. Note that the axes are illustrated in $\ln(|\Delta_t|)$ and $\ln(s)$.

Here we perform a local analysis at $\bar{t}$ near the surface intersection $t^*$, where $f(\mathbf{p}(t^*)) = 0, \bar{t} = t^* + \Delta_t$. And we let $\frac{\mathrm{d}f}{\mathrm{d}t}(\mathbf{p}(t^*)) = \mu$, and $\frac{\mathrm{d}^2 f}{\mathrm{d}t^2}(\mathbf{p}(t^*)) = \tau$. As a second-order analysis, we assume that in this local interval $t \in [t_l, t_r]$, $\frac{\mathrm{d}^2 f}{\mathrm{d}t^2}(\mathbf{p}(t))$ is fixed. After substitution and organization, the induced equation for local maximum point $\bar{t}$ is

$$\tau \cdot \left(1 + e^{-s(\mu\Delta_t + \frac{1}{2}\tau\Delta_t^2)}\right) = (\mu + \tau\Delta_t)^2 \cdot \left(s\left(1 - e^{-s(\mu\Delta_t + \frac{1}{2}\tau\Delta_t^2)}\right)\right), \tag{10}$$

which we will analyze later.

**The case of the naive solution**. Here we conduct a similar local analysis as in case of our solution. Regarding naive solution, when $w(t)$ attains local maximum at $\bar{t}$, there is:

$$(\mu + \tau\Delta_t) \cdot \left(-\left(1 - e^{-2s(\mu\Delta_t + \frac{1}{2}\tau\Delta_t^2)}\right)\right) = e^{-s(\mu\Delta_t + \frac{1}{2}\tau\Delta_t^2)}. \tag{11}$$

**Comparison.** Based on Eqn. 10 and Eqn. 11, we can numerically solve the equations on $\Delta_t$ for any given values of $\mu, \tau$, and $s$. Below we plot the curves of $\Delta_t$ versus increasing $s$ for different (fixed) values of $\mu, \tau$ in Fig. 1.

As shown in Fig. 1, the error of local maximum position $\Delta_t = O(s^{-2})$ for our solution and the error $\Delta_t = O(s^{-1})$ for the naive solution. That is to say, our error converges to zero faster than the error of the naive solution does as the standard deviation $1/s$ of the $S$-density approaches to 0, which is quadratic convergence versus linear convergence.

# D    Additional Experimental Details

## D.1    Additional Implemenation Details

**Network architecture**. We use a similar network architecture as IDR [10], which consists of two MLPs to encode SDF and color respectively. The signed distance function $f$ is modeled by an MLP that consists of 8 hidden layers with hidden size of 256. We replace original ReLU with Softplus with $\beta = 100$ as activation functions for all hidden layers. A skip connection [6] is used to connect the input with the output of the fourth layer. The function $c$ for color prediction is modeled by a MLP with 4 hidden layers with size of 256, which takes not only the spatial location $\mathbf{p}$ as inputs but also the view direction $\mathbf{v}$, the normal vector of SDF $\mathbf{n} = \nabla f(\mathbf{p})$, and a 256-dimensional feature vector from the SDF MLP. Positional encoding is applied to spatial location $\mathbf{p}$ with 6 frequencies and to view direction $\mathbf{v}$ with 4 frequencies. Same as IDR, we use weight normalization [7] to stabilize the training process.

**Training details**. We train our neural networks using the ADAM optimizer [3]. The learning rate is first linearly warmed up from 0 to $5 \times 10^{-4}$ in the first 5k iterations, and then controlled by the cosine decay schedule to the minimum learning rate of $2.5 \times 10^{-5}$. We train each model for 300k iterations for 14 hours (for the 'w/ mask' setting) and 16 hours (for the 'w/o mask' setting) in total on a single Nvidia 2080Ti GPU.

**Alpha and color computation**. In the implementation, we actually have two types of sampling points - the sampled section points $\mathbf{q}_i = \mathbf{o} + t_i\mathbf{v}$ and the sampled mid-points $\mathbf{p}_i = \mathbf{o} + \frac{t_i+t_{i+1}}{2}\mathbf{v}$, with section length $\delta_i = t_{i+1} - t_i$, as illustrated in Figure 2. To compute the alpha value $\alpha_i$, we use the section points, which is $\max(\frac{\Phi_s(f(\mathbf{q}_i)) - \Phi_s(f(\mathbf{q}_{i+1}))}{\Phi_s(f(\mathbf{q}_i))}, 0)$. To compute the color $c_i$, we use the color of the mid-point $\mathbf{p}_i$.

**Hierarchical sampling**. Specifically, we first uniformly sample 64 points along the ray, then we iteratively conduct importance sampling for $k = 4$ times. The coarse probability estimation in the i-th iteration is computed by a fixed $s$ value, which is set as $32 \times 2^i$. In each iteration, we additionally sample 16 points. Therefore, the total number of sampled points for NeuS is 128. For the 'w/o mask' setting, we sample extra 32 points outside the sphere. The outside scene is represented using NeRF++ [11].

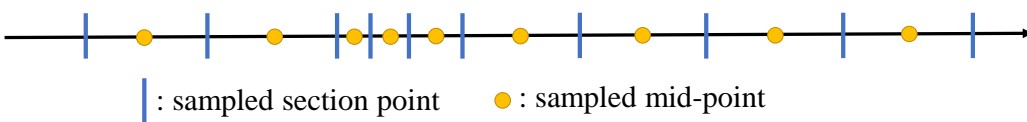

| : sampled section point    ● : sampled mid-point

Figure 2: The section points and mid-points defined on a ray.

| Scan ID | Threshold 0 | Threshold 25 | Threshold 50 | Threshold 100 | Threshold 500 |
|---------|-------------|--------------|--------------|---------------|---------------|
| Scan 40 | 2.36 | **1.79** | 1.86 | 2.07 | 4.26 |
| Scan 83 | 1.65 | **1.20** | 1.37 | 2.24 | 29.10 |
| Scan 114 | 1.62 | **1.04** | 1.10 | 1.43 | 8.66 |

Table 1: The Chamfer distances between the ground-truth and the level-set surfaces extracted from the NeRF results using different threshold values on three scenes from the DTU dataset.

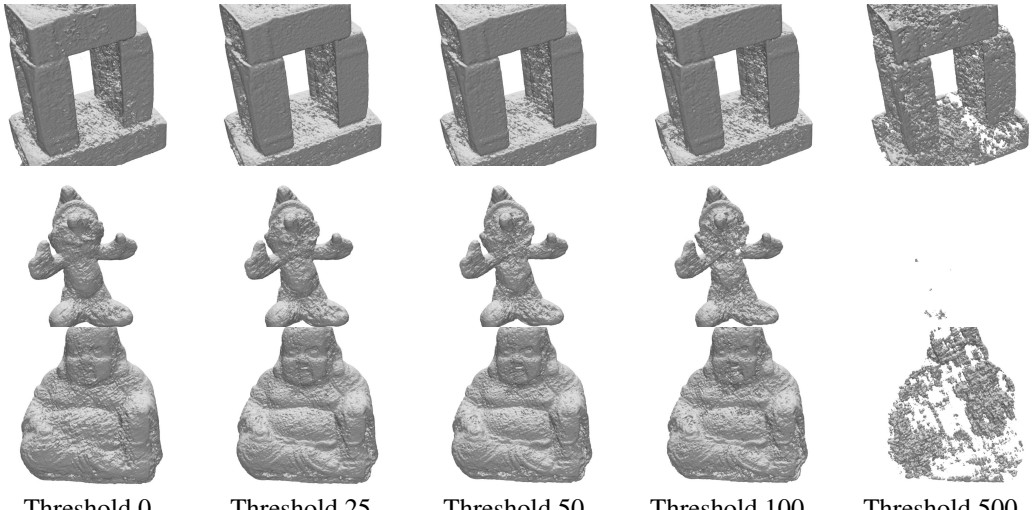

Figure 3: The visualization of the level-set surfaces extracted from the NeRF results using different threshold values.

| Threshold 0 | Threshold 25 | Threshold 50 | Threshold 100 | Threshold 500 |

## D.2 Baselines

**IDR[10]**. To implement IDR, we use their officially released codes[1] and pretrained models on the DTU dataset.

**NeRF[4]**. To implement NeRF, we use the code from nerf-pytorch[2]. To extract surfaces from NeRF, we use the density level-set of 25, which is validated by experiments to be the best level-set with smallest reconstruction errors, as shown in Table 1 and Figure 3.

**COLMAP[8]**. We use the officially provided CLI(command line interface) version of COLMAP. Dense point clouds are produced by sequentially running following commands: (1) *feature_extractor*, (2) *exhaustive_matcher*, (3) *patch_match_stereo*, and (4) *stereo_fusion*. Given dense point clouds, meshes are produced by (5) *poisson_mesher*.

**UNISURF[5]**. The quantitative and qualitative results in the paper are provided by the authors of UNISURF.

# E   Additional Experimental Results

## E.1   Rendering Quality and Speed

Besides the reconstructed surfaces, our method also renders high-quality images, as shown in Figure 4. Rendering an image in resolution of 1600x1200 costs about 320 seconds in the default volume rendering setting on a single Nvidia 2080Ti GPU. In addition, we also tested another sampling strategy by first applying sphere tracing to find the regions near the surfaces and only sampling points

[1]https://github.com/lioryariv/idr

[2]https://github.com/yenchenlin/nerf-pytorch

| Scan ID | 24 | 37 | 40 | 55 | 63 | 65 | 69 | 83 | 97 | 105 | 106 | 110 | 114 | 118 | 122 | Mean |
|---|---|---|---|---|---|---|---|---|---|---|---|---|---|---|---|---|
| PSNR(Ours) | 28.20 | 27.10 | 28.13 | 28.80 | 32.05 | 33.75 | 30.96 | 34.47 | 29.57 | 32.98 | 35.07 | 32.74 | 31.69 | 36.97 | 37.07 | 31.97 |
| PSNR(Ours$_{ST}$) | 27.07 | 26.58 | 27.70 | 28.37 | 31.32 | 31.39 | 30.20 | 31.79 | 28.58 | 30.87 | 33.61 | 32.40 | 31.33 | 35.55 | 35.96 | 30.85 |
| SSIM(Ours) | 0.764 | 0.813 | 0.737 | 0.768 | 0.917 | 0.835 | 0.845 | 0.850 | 0.837 | 0.837 | 0.875 | 0.876 | 0.861 | 0.891 | 0.892 | 0.840 |
| SSIM(Ours$_{ST}$) | 0.757 | 0.811 | 0.736 | 0.759 | 0.915 | 0.788 | 0.813 | 0.812 | 0.794 | 0.811 | 0.852 | 0.862 | 0.847 | 0.867 | 0.873 | 0.820 |

Table 2: Quantitative comparisons by different sampling strategies. -$ST$ indicates the sampling strategy with sphere tracing.

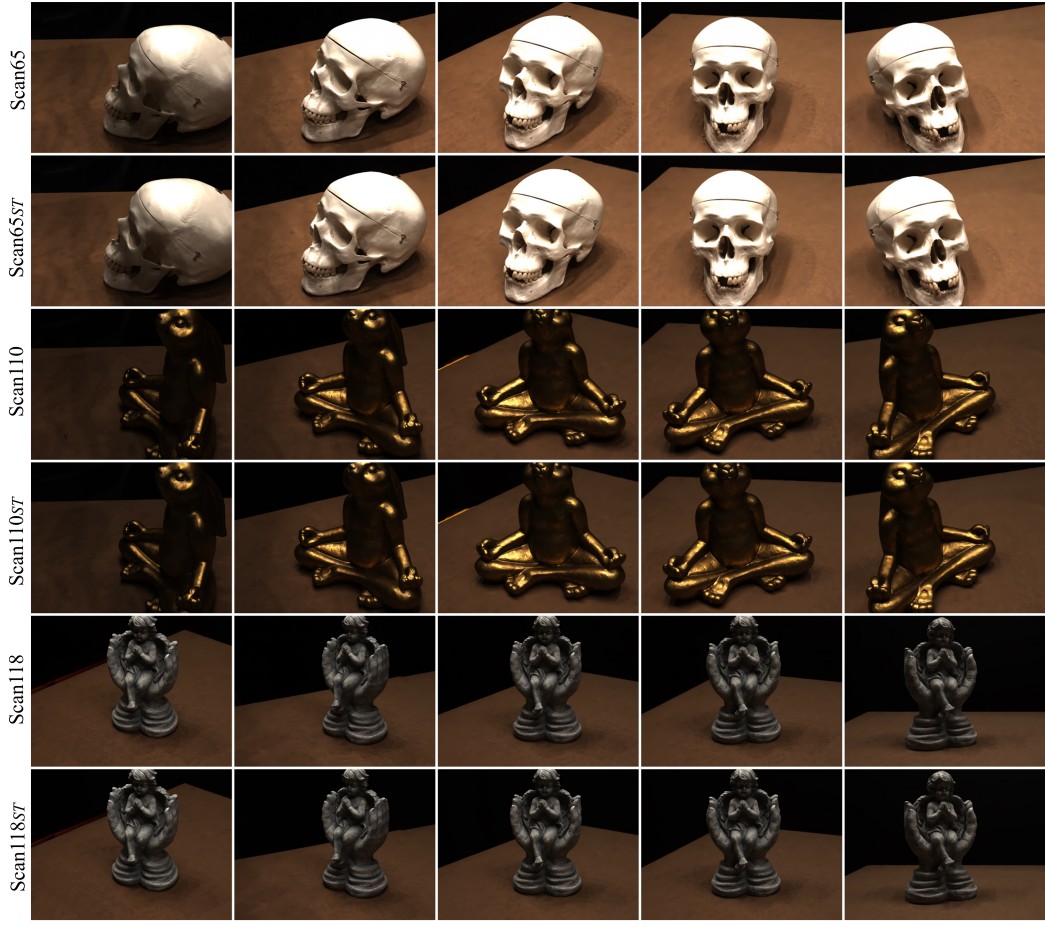

Figure 4: Rendered images by our method on the DTU dataset using different sampling strategies. -$ST$ indicates the sampling strategy using sphere tracing.

| Scan ID | 24 | 37 | 40 | 55 | 63 | 65 | 69 | 83 | 97 | 105 | 106 | 110 | 114 | 118 | 122 | Mean |
|---|---|---|---|---|---|---|---|---|---|---|---|---|---|---|---|---|
| PSNR(NeRF) | 24.83 | 25.35 | 26.87 | 27.64 | 30.24 | 29.65 | 28.03 | 28.94 | 26.76 | 29.61 | 32.85 | 31.00 | 29.94 | 34.28 | 33.69 | 29.31 |
| PSNR(Ours) | 23.98 | 22.79 | 25.21 | 26.03 | 28.32 | 29.80 | 27.45 | 28.89 | 26.03 | 28.93 | 32.47 | 30.78 | 29.37 | 34.23 | 33.95 | 28.55 |
| SSIM(NeRF) | 0.753 | 0.794 | 0.780 | 0.761 | 0.915 | 0.805 | 0.803 | 0.822 | 0.804 | 0.815 | 0.870 | 0.857 | 0.848 | 0.880 | 0.879 | 0.826 |
| SSIM(Ours) | 0.732 | 0.778 | 0.722 | 0.739 | 0.915 | 0.809 | 0.818 | 0.831 | 0.812 | 0.815 | 0.866 | 0.863 | 0.847 | 0.878 | 0.878 | 0.820 |

Table 3: Quantitative comparisons with NeRF on the task of novel view synthesis without mask supervision.

in those regions. With this strategy, rendering an image in the same resolution only needs about 60 seconds. Table 2 reports the quantitative results in terms of PSNR and SSIM in default volume rendering setting and sphere tracing setting.

## E.2 Novel View Synthesis

In this experiment, we held out 10% of the images in the DTU dataset as the testing set and the others as the training set. We compare the quantitative results on the testing set in terms of PSNR and SSIM with NeRF. As shown in Table 3, our method achieves comparable performance to NeRF.

### E.3 SDF Qualitative Evaluation

While our method without Eikonal regularization [2] or geometric initialization [1] produces plausible surface reconstruction results, our full model can predict a more accurate signed distance function as shown in Figure 5. Furthermore, using random initialization produces axis-aligned artifacts due to the spectral bias of positional encoding [9] while the geometric initialization [1] does not have such kind of artifacts.

### E.4 Training Progression

We show the reconstructed surfaces at different training stages of the Durian in the BlendedMVS dataset. As illustrated in Figure 6, the surface gets sharper along the training process. Meanwhile, we also provide a curve in the figure to show how the trainable standard deviation in $\phi_s$ changes in the training process. As we can see, the optimization process will automatically reduce the standard deviation so that the surface becomes more clear and sharper with more training steps.

### E.5 Limitation

Figure 7 shows a failure case where our method fails to correctly reconstruct the texutreless region of the surface on the metal rabbit model. The reason is that such textureless regions are ambiguous for reconstruction in neural rendering.

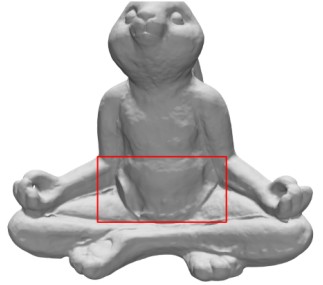
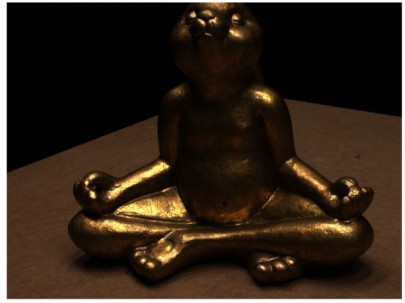

Figure 7: A failure reconstruction case containing textureless regions.

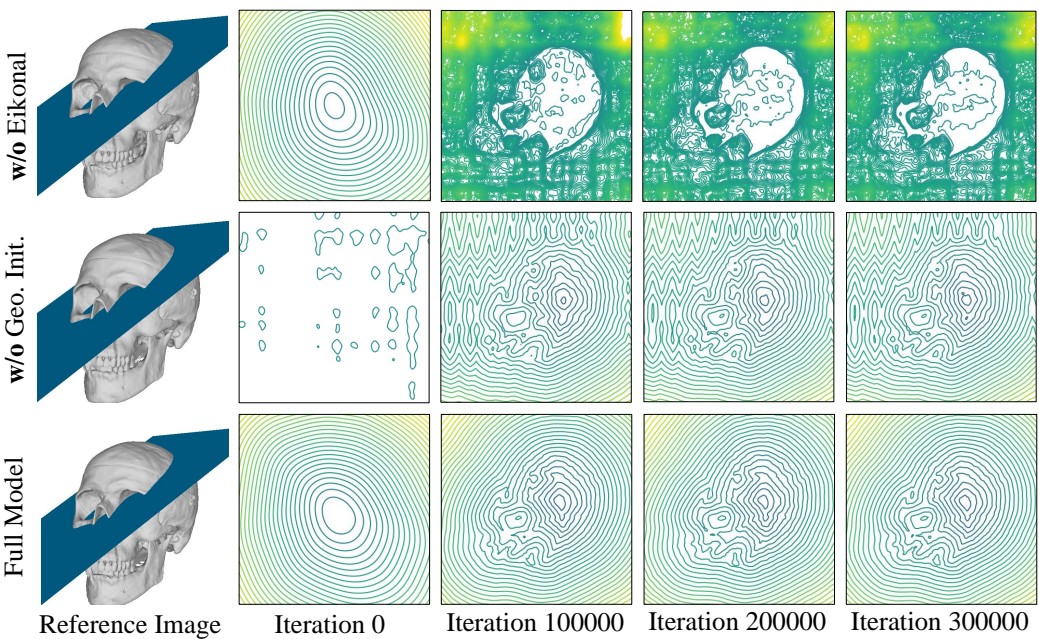

Figure 5: Visualization of signed distance fields on the cutting plane (blue plane of the left image) in different training iterations.

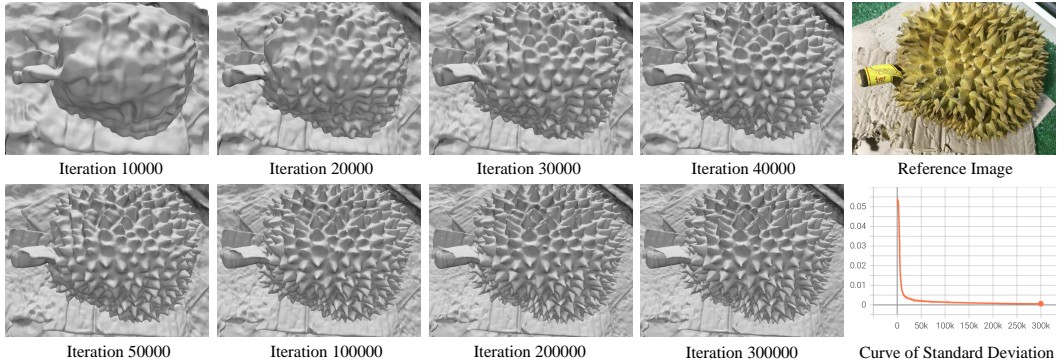

Figure 6: Training progression of the Durian in the BlendedMVS dataset. The bottom right figure shows the curve of the trainable standard deviation in the training progress.

## E.6 Additional Results

In this section, we show additional qualitative results
on the DTU dataset and BlendedMVS dataset. Figure 8 shows the comparisons with baseline methods in both **w/** mask setting and **w/o** mask setting. Figure 9 shows addtional results in **w/o** mask setting.

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

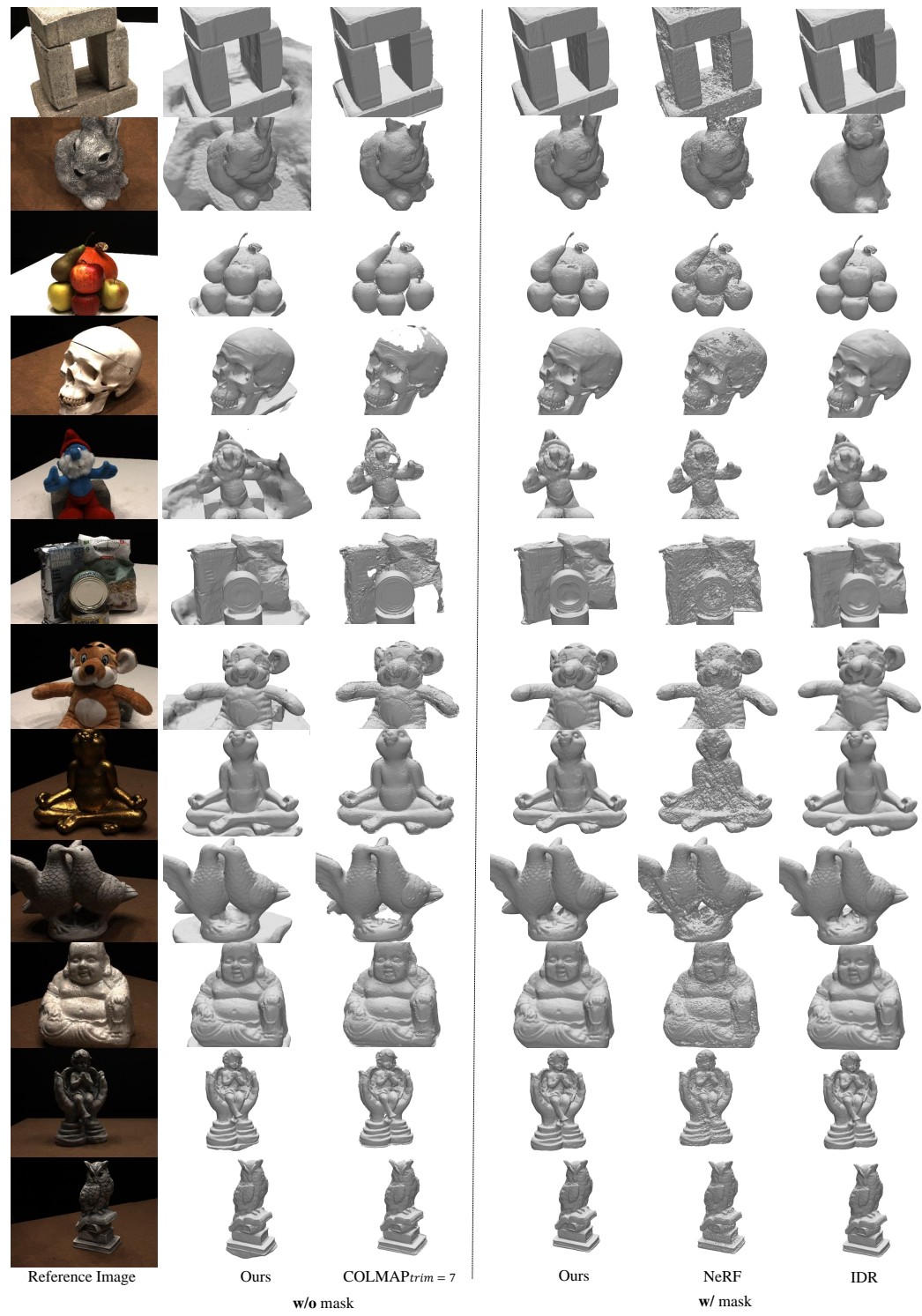

| Reference Image | Ours | COLMAP$_{trim=7}$ | | Ours | NeRF | IDR |
|---|---|---|---|---|---|---|
| | | **w/o** mask | | | **w/** mask | |

Figure 8: Additional reconstruction results on the DTU dataset.

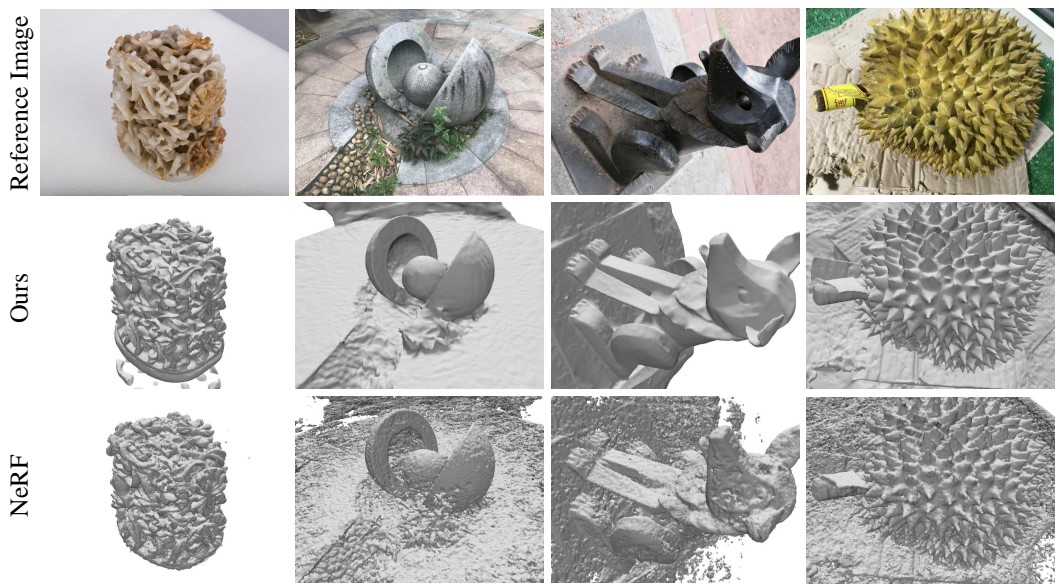

Figure 9: Additional reconstruction results on BlendedMVS dataset without mask supervision.