# OpenReview forum: "NeuS: Learning Neural Implicit Surfaces by Volume Rendering for Multi-view Reconstruction"
_NeurIPS.cc/2021/Conference — NeurIPS 2021 Spotlight_

### Official Review · Reviewer_MRq8 · 2021-07-12

**Rating:** 8
**Confidence:** 4

**Summary:**

This paper proposes a neural surface reconstruction method which they call called NeuS. The novelty of this method is derived from their use of signed distance functions to represent geometry, and a proposed volume rendering method for accumulating information along rays. With these changes they are better able to estimate depth in more complicated surface geometry, demonstrate clearly superior shape predictions, and achieve state of the art quantitative results on the DTU dataset.

**Ethical Concerns:**

No.

**Limitations And Societal Impact:**

Yes.


**Main Review:**

This paper is quite strong. It identified an issue with NERF style volumetric rendering method, and solved  it by representing geometry with an SDF. The proposed volume rendering method which allows for this change is well motivated and explained. Both the qualitative and quantitative results are convincing and compelling.

One point on concern is the comparison to other methods. I am not an expert in this field but it seems there has been quite an explosion of nerf-like papers since its release, and from a quick lit review of them it seems like NERF would no longer be considered state of the art in novel view synthesis, as referenced in the baselines section. NERF was released only last year and so there is leeway here for concurrent, or close to concurrent research, but I would perhaps revise this claim. If you wanted to raise my score higher it might be nice to compare directly to papers like "Neural Sparse Voxel Fields" which I see you reference, or other more recent papers which seem to tackle the same shortcomings in the original paper.

other comments:
- lines 67 to 70 are true of some methods but not all. This overly general statement is misleading.
- There needs to be some editing for grammar, especially in the intro section.


**Time Spent Reviewing:**

2

---

> ### Author Response · Authors · 2021-08-10
> **Response**
>
> We thank the reviewer for the detailed comments and constructive suggestions. Below are our responses to the questions.
>
> **R4-Q1. Comparison of state-of-the-art of neural volume rendering.**
> The volume density based methods, including NeRF and NSVF, have the same difficulty in extracting high-quality surfaces since they lack constraints on surface geometry. We therefore chose a typical volume density based method, NeRF, as our baseline. The only neural volume rendering method that attempts to address the issue of surface extraction is the concurrent work, UNISURF, which we compared with in the submission.
>
> **R4-Q2. Clarity of related works review & Typos.** Thank you for pointing out these issues. We will improve the review of related works and fix the typos in the revision accordingly.

---

### Official Review · Reviewer_Hvd1 · 2021-07-17

**Rating:** 8
**Confidence:** 5

**Summary:**

This paper presents a method for implicit 3D surface reconstruction from posed 2D images, where the 3D surface is represented with an SDF. The authors advocate a new way of using neural volume rendering methods for surface reconstruction, where the density field is induced by the optimized SDF instead of direct MLP outputs (as in e.g. NeRF). In addition, the authors described two key properties that should be satisfied for volume rendering with SDF representations and proposed a novel solution. Experimental results on the DTU multiview dataset shows its advantage over state-of-the-art baseline methods.

**Limitations And Societal Impact:**

Adequately addressed

**Main Review:**

Strengths:
+ The paper is very well-written, with a clear problem statement and a thorough overview of related literature.
+ To take advantage of the nice optimization properties in neural volume rendering methods e.g. NeRF, the authors pointed out two important properties which should be satisfied for extracting surfaces with SDF as the representation: unbiasedness and occlusion-awareness. The authors suggested two simple solutions satisfying only each of the property, and then propose a nice solution to blend the two together. This makes a complicated concept easy to understand. The authors also provided a nice theoretical proof on how the proposed solution satisfy both properties.
+ The proposed method NeuS is compared with extensive recent baseline methods and significantly outperforms both qualitatively and quantitatively. Ablation studies and failure cases are also provided, which helps understand the contribution of each component.

Weaknesses:
- It was mentioned that a proof on how Eq. 6 is not unbiased would be provided, but only an empirical example was provided in the supplementary materials. It is still unclear to me how the construction of Eq. 6 is not unbiased.
- Although the results are superior, it would be helpful to provide a comparison on the training/inference speed. Since NeuS is still a volume rendering method with hierarchical sampling as adopted in NeRF, I would expect it to be almost as slow as NeRF to render. How slow would NeuS be compared to IDR?
- It would be good to also evaluate how well the recovered 3D representation actually satisfy an SDF, as SDF recovery is also one main goal of the paper. This would be important since efficient rendering (e.g. with sphere tracing) and relighting could be applied on true SDF shapes. The current evaluation seem to focus on 3D shape reconstruction in terms of surface accuracy (measured by Chamfer distances). In Fig. 4 of the supplementary material, the exterior region does not seem to correctly satisfy an SDF.

I think it would also be interesting to see whether Neus would be able to reconstruct surfaces on forward-facing scenes, such as in the LLFF dataset originally considered in NeRF. Does the reparametrization of coordinates using NDC affect the solution?

Other minor problems:
- It is unclear what $s$ in L123 -- is it a hyperparameter that should be tuned?
- $\delta_i$ is undefined in Eq. 6. (I'm guessing it corresponds to $t_{i+1}-t_i$?)
- L183: the equation seems off, as $\bar{\phi} = \phi_s \circ f$ (Eq. 2).
- L192: the differential is missing.

**Time Spent Reviewing:**

4hr

---

> ### Author Response · Authors · 2021-08-10
> **Response**
>
> We thank the reviewer for the detailed comments and constructive suggestions. Below are our responses to the questions.
>
> **R3-Q1. Demonstration of biased problems in Naive Solution II is empirical.**  We will provide a formal proof in a continual form in the revision.
> Here we provide a brief sketch of our proof: By taking the derivative of the weight function $w(t) = T(t)\sigma(t)$, we get $\frac{{\rm d} w}{{\rm d} t} = T(t)(\frac{{\rm d} \sigma(t)}{{\rm d} t} - \sigma(t)^2)$. Checking its sign with the adoption of Naive Solution II that $\sigma(t)=\phi_s(f(\mathbf{p}(t)))$ at the surface intersection $t^*$ where $f(\mathbf{p}(t^*))=0$, we find that the derivative is less than zero, which shows the weight term does not attain its maximum at surface intersection if Naive Solution II is used. The proof still holds true even though we assume locally the surface is a planer patch.
>
> **R3-Q2. Training & inferencing time.** The response is in `All-Q3`.
>
> **R3-Q3. Evaluation of SDF quality.** The response is in `All-Q2`.
>
> **R3-Q4. Test on forward-facing dataset.**  Theoretically, our method should also work on forward-facing scenes if they can be well-bounded inside a sphere. But the scenes in LLFF dataset are mostly not well-bounded and should be reparameterized to an NDC space for better scene representation. Since our method is based on SDF in the Euclidean space, how to properly define an SDF in the NDC space still remains an open problem. It would be an interesting future work to reconstruct surfaces in reparameterized spaces.
>
> **R3-Q5. It is unclear what $s$ is in L123 -- is it a hyperparameter that should be tuned?**  As described in section 4.2 of the supplementary material, $s$ is a learnable parameter that is jointly optimized in the training process. We will make the description clearer in the method section.
>
> **R3-Q6. Typos.** Thank you for pointing out the typos. We will fix them in the revision accordingly.

---

### Official Review · Reviewer_3fHi · 2021-07-17

**Rating:** 8
**Confidence:** 4

**Summary:**

This paper presents a novel multi-view reconstruction model that couples an implicit SDF representation (as in IDR [31]) with an unbiased volumetric rendering function (as in NeRF [20]). The volumetric rendering function is deliberately "de-biased" by redefining the opacity values $\alpha_i$ such that the weight is maximized exactly on the surface defined by the zero level set of the SDF.

The resulting model enjoys the benefits of both surface-based representation for high-fidelity surface reconstruction, and dense volumetric rendering which provides effective gradients that are not restricted locally to the surface and facilitate surface details. The reconstructed surfaces are much better compared to IDR and NeRF. This result can potentially also be very useful for further downstream applications that require smooth effective gradient signals.

**Ethical Concerns:**

I do not see immediate concerns.

**Limitations And Societal Impact:**

A brief example of slightly inaccurate reconstructions is provided in the supplementary material. Similar imperfections are also visible in other examples, such as in purely specular or saturated regions. But I wonder whether there are more extreme major cases.


**Main Review:**

## Strengths
### S1 - method
The idea of combining volumetric rendering with SDF is actually very neat, which is better than the analytical gradient used in DVR and IDR that is only defined on surface and hence very local. This is very well implemented by carefully designing the alpha values to observe the surface properties of the SDF, driven by mathematical proofs. The resulting model is clean and easily interpretable.

### S2 - results
The reconstructed surfaces are clearly better than existing methods, eg IDR, NeRF, UNISURF, with smooth surfaces (unlike noisy shapes extracted from NeRF) and fine details, handling thin structures well (much better than IDR). The model can also perform reconstruction without object masks without much loss in quality, whereas DVR and IDR require mask supervision.

Thorough ablation and analysis results, including visualizations of the training progression, are presented and provide a lot of insights to the model.

### S3 - writing
The paper is very well written and highly polished. I really enjoyed reading it. The motivations are clearly explained and illustrated with examples and great figures. Technical descriptions are followed by motivation/intuition and careful proofs, and very well structured such that the readers could easily follow through.


## Weaknesses
### W1 - image synthesis
There are no comparison on the image synthesis quality. It seems the rendered image appear less visually appealing compared to NeRF, which however is a trade-off for obtaining explicit surface.

### W2 - (minor comment) more applications
Although the paper is already very complete in itself, it mostly follows the experimental setup of previous method (which is good). There are also many other interesting applications now that the model can obtain smooth, non-local gradients with volumetric rendering while maintaining a surface representation. For example, does that helps with the speed of convergence? Have the authors tried optimizing camera poses from coarse estimates as IDR does, and would it tolerate more noise? Does it help with the cases with fewer views?

## Typos
- Line 176 & 183: $\bar{\phi}$ should be $\phi$
- Line 152: remove "{"
- Line 162: "a unbiased" -> "an unbiased"


---
## Post-Rebuttal
I appreciate the authors' efforts in the rebuttal. This is a solid submission with a novel method and great results. I will keep my original rating.

**Time Spent Reviewing:**

4

---

> ### Author Response · Authors · 2021-08-10
> **Response**
>
> We thank the reviewer for the detailed comments and constructive suggestions. Below are our responses to the questions.
>
> **R2-Q1. Novel view synthesis.** See the answer in `All-Q1`.
>
> **R2-Q2.  Camera pose refinement, help with the convergence speed, fewer view inputs, etc.**
> Our model can reconstruct high-quality surfaces on the DTU dataset and BlendedMVS dataset where there exist noises in the camera poses, without camera pose refinement. However, camera pose refinement would help when there are significant camera noises.
> We did not observe that our model can help with the cases with fewer views or the convergence of training.
>
> **R2-Q3. Typos.** Thank you for pointing out the typos. We will fix them in the revision accordingly.

---

### Official Review · Reviewer_f9A2 · 2021-07-19

**Rating:** 7
**Confidence:** 4

**Summary:**

This paper proposes a method that combines volume rendering with surface rendering techniques for multi-view reconstruction. The method represents the geometry of objects using a neural implicit representation that outputs the SDF at query locations. In order to enable volume rendering, the output SDF is mapped to a density function centered around the surface. Furthermore, they also encode the appearance of the object into the neural implicit representation that is needed for multi-view reconstruction. The main contribution is a novel weight function that is centered (unbiased) around the surface while being occlusion-aware. Therefore, the paper resolves major limitations of volume rendering (to few constraints on surface) and surface rendering (very sparse gradients). The paper extensively discusses different formulations of the weight function (properties, weaknesses, strengths). The method is evaluated on the DTU as well as the BlendedMVS dataset and compared to existing state-of-the-art methods like NeRF (volume rendering) and IDR (surface rendering). Moreover, they evaluate some aspects of the method in an ablation study.


**Limitations And Societal Impact:**

They are discussed sufficiently.

**Main Review:**

Here, I order the strengths and weaknesses according to their importance and contribution to the final score as well as additional comments that should be resolved in the final version of the paper.

#Strengths:

1. The paper proposes a simple yet elegant formulation for multi-view surface reconstruction using neural implicit representations. The results are impressive and show that it is very beneficial to combine volume rendering and surface rendering to solve the well known problems of noisy surfaces (NeRF) and requiring a mask for supervision (IDR). This is particularly shown in Figure 6.

2. The paper extensively discusses the strengths and limitations of existing weight functions used in volumetric rendering. Especially, the illustrations in Figure 2 and Figure 3 are very useful to understand the problem and the proposed solution.

3. The introduction as well as the related work very well motivate the paper with respect to existing surface and volume rendering methods. This helps the reader to understand the importance of the problem and the proposed solution.

4. By providing an intuition behind different weight functions, the different solutions and their strengths/weaknesses are very well motivated.

#Weaknesses:

1. Clarity:
The paragraph, where the solution is explained (L185 - L193) is very unclear. It is very hard to understand how the solution is derived from Equation 5 and Equation 6. This should be definitely made clearer for the final version of the paper as it is crucial for the reader to understand the derivation of the main contribution. I would like to see how you derive and combine equation 7 from equation 5 and 6. This could then be also added to the supplementary material for the final paper with an intuition in the main paper.

2. Results/Ablations

- Firstly, as you also encode the appearance and especially NeRF is primarily designed for novel view synthesis, it would be interesting to see how this method performs on novel view synthesis. For doing this, I suggest to visualize novel views from the scenes and also run a quantitative evaluation using PSNR and SSIM.

- Secondly, as you already compare to UNISURF, it would also be interesting to see the experiments on the Indoor Scene Dataset as shown in UNISURF (Figure 8, p. 8).

- Thirdly, one missing experiment is the ablation of all loss terms. I would be particularly interested in an ablation of the Eikonal term to see whether this is also the reason for the better performance compared to UNISURF.


3. Related Work:
There are some relevant works missing on multi-view reconstruction. Particularly, the two works below that are one of the first works using machine learning for multi-view reconstruction.

- Kar et al., Learning a multi-view stereo machine, NeurIPS 2017
- Choy et al., 3D-R2N2: A Unified Approach for Single and Multi-view 3D Object Reconstruction, ECCV 2016

In the section about Neural Implicit Representations, the following work is missing:

- Sitzmann et al., DeepVoxels: Learning Persistent 3D Feature Embeddings, CVPR 2019

Moreover, I do only partially agree with the discussion of the differences between this work and the concurrent work UNISURF. Although it is correct that UNSIRF represents the geometry as occupancy values, you can also naturally extract the surface at the decision boundary (0.5), which is just a different level set. I doubt that this difference is the reason for the performance improvement over UNISURF. A more in depth discussion would be helpful for the community to understand what improves performance and what does not.

4. Lacking justification of certain design choices:

- To me, the choice of the logistic density distribution is not motivated enough. I assume the main benefit is the easy integration to the sigmoid function that is used in the final solution (Equation 7). I would like to see a brief comment on that.


5. Clarity/Flow of Reading
The clarity of the writing and flow of reading is limited by very long sentences (e.g. Abstract L11 - L14), obscure formulations (e.g. L87 'which makes the gradient only be backpropagated to a local region near the intersection', L118 'neural networks of Multi-layer Perceptron (MLP), L), and wrong/inconsistent prepositions (e.g. requirements of weight function (L147-L148) vs. requirements on weight function (L149)). This significantly breaks the flow while reading the paper and makes it harder to understand everything.


#Additional Comments:

- The writing of the paper should be reviewed and revised in general (uncountable vs. countable nouns, third person s, prepositions, missing or too many a/the (e.g. L56, L86), etc.). This would make the reading much easier and redirects the mental capacity of the reader to the content.

- A few formulations could be improved for better clarity.
	- L31 - L32: would get optimization stuck --> other formulation
	- L100: sample region --> sampling region
	- L108: zero-set --> zero level set
	- L111: train the network of SDF --> other formulation
	- L120: method to training the SDF network --> method to train the SDF network
	- L134: To learn the parameters of MLPS of the SDF and the color field --> To learn the parameters of the SDF and color networks (or something similar)
	- L157 - 158: near the view point --> closer to the view point
	- Theorem 1: depth --> depth values
	- Theorem 1: going from the outside of the surface --> going from the outside of the object (there is no inside outside of a surface)

- It would be very good to visualize the meshes (especially in Figure 7) with some sort of error color coding (similar to Tatarchenko et al. 2019, Figure 12). This would guide the reader to better see where the different configuration fail/work.

- Reference 5 and 6 are the same. They should be merged.

- The failure case in the appendix is not really informative (Fig. 6). Some more failure cases caused by lack of texture should be shown together with the input views (to see the lack of texture). Guiding the reader more (directly in the figure) would help as well.




#Questions

- What is the reason for using different sampling points for the color and alpha values? (Appendix L57-59)
- Why do the results for some scenes get worse when using the mask supervision? What is the intuition behind that? (Table 1)


**Time Spent Reviewing:**

8

---

> ### Author Response · Authors · 2021-08-10
> **Response**
>
> We thank the reviewer for the detailed comments and constructive suggestions. Below are our responses to the questions.
>
> **R1-Q1. Clarity of weight deviation.**
> We will revise this paragraph to make the deviation of our solution clearer. In the paper, we directly gave the formula of $\alpha(t_i)$. Here, to help the understanding, we briefly introduce the idea of how we derive the proposed unbiased $\alpha(t_i)$. We will provide a complete derivation in our revised paper.
>
> To derive the expected weight function, we first consider a simple case where there is only a single plane intersection along the ray. There is no occlusion in this case, so we let our weight be the same as the weight defined in Naive Solution I here. Then following the classic volume rendering framework that $w(t) = \sigma(t)T(t), T(t)=\exp(-\int_{0}^{t}\sigma(u){\rm d}u)$, we derive the underlying volume density $\sigma$ in this case.  We then generalize this density to the general cases with multiple surfaces intersection. As a result, the expected volume density in general cases can be derived by
> $$\sigma(t) = \max(\frac{-\frac{{\rm d}\Phi_s}{{\rm d}t}(f(\mathbf{p}(t)))}{\Phi_s(f(\mathbf{p}(t)))}, 0){\rm .}$$
>
> The value $\alpha(t_i)$ in Eqn. 7 of the paper is a discrete form of $\sigma(t)$ by computing $\alpha(t_i) = 1 - \exp(-\int_{t_{i}}^{t_{i+1}}\sigma(t){\rm d} t)$. This gives the derivation of our alpha value in the paper.
>
> **R1-Q2. Novel view synthesis.** See the answer in `All-Q1`.
>
> **R1-Q3. Indoor scene reconstruction.**  We tested our method and NeRF on one scene from the SceneNet dataset. Our quantitative result in terms of Chamfer distance to the ground-truth is 0.0165, which is better than that of NeRF (0.0256). We will add the experiments on SceneNet to the revision.
>
> **R1-Q4. Ablation study of Eikonal term.** The Eikonal term is used to enforce the predicted scene representation to be an SDF. We conducted an experiment without the Eikonal loss term on the skull model (DTU scan65). Surprisingly, the result in terms of Chamfer distance (0.68) without the Eikonal term is on par with our full model design (0.72). We think it is because, even without the Eikonal constraint, the unbiased property of our weight design is still satisfied. Thus, our method without the Eikonal constraint is still able to predict accurate surfaces (i.e. the zero level set of the SDF), but is not capable of reconstructing all level sets of the SDF accurately. As reported in `All-Q2`, by directly computing the errors for the predicted SDF and the ground-truth SDF across all the level sets, the mean absolute errors (MAE) of our method w/ and w/o the Eikonal constraint are 1.30 and 211.53 respectively, demonstrating that the quality of the predicted SDF has been improved significantly with the Eikonal constraint.
>
> **R1-Q5. Choice of logistic density function.**  As explained in L122, theoretically, the density function used in our framework can be any unimodal density function. We chose the logistic distribution because the CDF of logistic distribution can be analytically computed with a Sigmoid function, while a commonly-used Gaussian distribution has no analytical equation for CDF.
>
> **R1-Q6. Why use different types of sampling points for alpha and color calculation.** The reason is that the value $\alpha_i$ is defined on a line section, which represents the opacity of the whole section. Consequently, to represent the color of the sampled section, we chose the color at the midpoint color rather than that at the starting point or end point.
>
> **R1-Q7. Why do the results for some scenes get worse when using the mask  supervision?**
> Among the 15 scenes in DTU dataset, there are two cases (scan 40, 63) where the quantitative results with mask supervision are observably worse than those without mask supervision. As shown in Fig. 10 of the supplementary material, in row 1 (scan 40) and row 3 (scan 63) the results with mask supervision have more incorrect concave surfaces on textureless regions than those without mask supervision. We speculate that it is because imposing mask loss encourages the surface to shrink to coincide with the mask, which results in concave surfaces on textureless regions.
>
> **R1-Q8. Discussion of the differences to UNISURF is not totally correct.**   We agree that 0.5 is also a valid decision boundary to define the surfaces of occupancy fields, and UNISURF also successfully applies volume rendering to reconstruct high-accuracy surfaces from the input views. However, our point is that this decision boundary used in UNISURF is empirical and the weight is not guaranteed to be unbiased. In contrast, our method is proved unbiased and thus outperforms UNISURF.
>
> **R1-Q9. Missing references, awkward flows & typos.** Thank you for pointing out these issues. We will add the missing references, improve the writing and fix the typos in the revision accordingly.

---

### Author Response · Authors · 2021-08-10
**Response**

We thank the reviewers for the constructive comments. Below are our responses to those common questions.

**All-Q1. Novel view synthesis.**
The goal of this work is to accurately reconstruct the surface geometry given multiple images as supervision. As suggested, we also conducted an evaluation of novel view synthesis on the DTU dataset, where we held out 7 or 8 images for testing from the full set of 49 or 64 images. The average scores (PSNR || SSIM) of our method on the test sets of all 15 scenes are 28.55 || 0.820, comparable to NeRF’s results: 29.31 || 0.826.

**All-Q2. Evaluation of SDF quality.**
As suggested, we evaluated the quality of the SDF predicted by our method on the skull model (DTU scan65). We uniformly sampled points in the bounding sphere of the object and computed the mean absolute error (MAE) between the signed distance from the predicted SDF and the ground-truth SDF. The MAE is 1.30 in world scale, which shows our predicted SDF is very close to the ground truth, and is significantly better than those without geometric initialization (16.07) and without Eikonal regularization (211.53).

**All-Q3. Training & inferencing time.**
As described in Section 4.1, the training time of each scene is around 14 hours (w/ background modeled by NeRF++) or 16 hours (w/o background) for 300k iterations. At inference time, rendering an image at the resolution of 1600x1200 takes around 320 seconds(w/ background modeled by NeRF++) or 250 seconds(w/o background).

We also tested a new sampling strategy by first applying sphere tracing to find the regions near the surfaces and only sampling points in those regions. With this strategy, rendering an image at 1600x1200 pixels only needs 60 seconds(w/ background modeled by NeRF++) or 30 seconds(w/o background), which is comparable to that by IDR (30 seconds per image, w/o background). Another acceleration strategy is to incorporate the sparse voxel structures as done in NSVF and PlenOctree. We will add this discussion to the revision.

---

### Decision · Program_Chairs · 2021-09-27

**Decision:**

Accept (Spotlight)

**Comment:**

The paper presented a novel idea of combining volumetric rendering with 3D surface representation (SDF) and achieved SOTA results. All the reviewers liked the paper and gave very constructive comments. Please include the results on novel view synthesis, evaluation of SDF quality, details of training  & inference time in the final version. Also, please carefully proof read the paper.